# IODVA1, a guanidinobenzimidazole derivative, targets Rac activity and Ras-driven cancer models

Anjelika Gasilina[1,2☯], Gurdat Premnauth[3☯], Purujit Gurjar[3], Jacek Biesiada[2,4], Shailaja Hegde[1,2], David Milewski[2,5], Gang Ma[1,2], Tanya V. Kalin[2,5], Edward Merino[3], Jarek Meller[2,4], William Seibel[2,6], José A. Cancelas[1,2,7], Lisa Privette Vinnedge[2,6], Nicolas N. Nassar[1,2]*

**1** Division of Experimental Hematology and Cancer Biology, Children's Hospital Research Foundation, Cincinnati, Ohio, Unites States of America, **2** Department of Pediatrics, University of Cincinnati College of Medicine, Cincinnati, Ohio, Unites States of America, **3** Department of Chemistry, University of Cincinnati, Cincinnati, Ohio, Unites States of America, **4** Division of Biomedical Informatics, Cincinnati Children's Hospital Medical Center, Cincinnati, Ohio, Unites States of America, **5** Division of Pulmonary Biology, Perinatal Institute, Cincinnati Children's Hospital Research Foundation, Cincinnati, Ohio, Unites States of America, **6** Division of Oncology, Cincinnati Children's Hospital Medical Center, Cincinnati, Ohio, Unites States of America, **7** Hoxworth Blood Center, University of Cincinnati Academic Health Center, Cincinnati, Ohio, Unites States of America

☯ These authors contributed equally to this work.
* nicolas.nassar@cchmc.org

**Data Availability Statement:** All relevant data are within the manuscript and its Supporting Information files.

## Abstract

We report the synthesis and preliminary characterization of IODVA1, a potent small molecule that is active in xenograft mouse models of Ras-driven lung and breast cancers. In an effort to inhibit oncogenic Ras signaling, we combined *in silico* screening with inhibition of proliferation and colony formation of Ras-driven cells. NSC124205 fulfilled all criteria. HPLC analysis revealed that NSC124205 was a mixture of at least three compounds, from which IODVA1 was determined to be the active component. IODVA1 decreased 2D and 3D cell proliferation, cell spreading and ruffle and lamellipodia formation through downregulation of Rac activity. IODVA1 significantly impaired xenograft tumor growth of Ras-driven cancer cells with no observable toxicity. Immuno-histochemistry analysis of tumor sections suggests that cell death occurs by increased apoptosis. Our data suggest that IODVA1 targets Rac signaling to induce death of Ras-transformed cells. Therefore, IODVA1 holds promise as an anti-tumor therapeutic agent.

## Introduction

With the increasing wealth in 3D structural information of biological targets, docking- or structure-based screening (also known as target-based drug design) is becoming the go-to technique to identify small molecules that bind to a specific pocket on a given biomolecular target to induce a specific biological outcome. In such a screen, a library of small molecules is docked computationally by exploring the conformational space of each tested compound in a docking program against a chosen pocket on a receptor's surface. Top compounds, ranked using a scoring function [1–3], are then tested in *in vitro* and cellular assays for the desired

**Funding:** This project was funded by NIH grants (R01CA115611 and R37CA218072 to N.N.N. and LPV, respectively), a Translational Research Program from the Leukemia Lymphoma Society (6076-14; to N.N.N. and J.A.C.), an Affinity Group Award from the Cincinnati Cancer Center (N.N.N and J.A.C), and a FY 19 Cincinnati Children's Innovation Fund award (N.N.N and J.A.C). The Funders had no role in study design, data collection and analysis, decision to publish, or preparation of the manuscript.

**Competing interests:** The authors have declared that no competing interests exist.

effect and selected hits are optimized by medicinal chemistry against their target, before ultimately testing the most efficient non-toxic lead(s) in a mouse model of disease for *in vivo* efficacy. An alternative approach, prevalent prior to popularization of target-based screening and still used today, is a phenotypic screen. In this screen, compound's effectiveness is evaluated through assays such as cell toxicity and proliferation prior to elucidating the compound's mechanism of action. Although each with their own advantages and disadvantages, both screens proved to be successful, with the phenotypic screens yielding more "first-in-class" and target-based screens yielding more "best-in-class" drugs, reviewed in [4].

Based on our longstanding interest in the small GTP-binding protein Ras signaling, we used virtual screening to identify small molecules that bind to Ras with the ultimate goal of reducing its signaling in disease. Ras signaling is tightly regulated by cycling between the inactive GDP-bound form and the active GTP-bound form. When active, Ras binds to a plethora of downstream effectors including Raf-kinase and PI-3 kinase (PI3K) to regulate, among others, cell growth, gene expression, and remodeling of the actin cytoskeleton [5, 6]. Activating mutations, upregulation of cell surface receptor signaling, or loss of negative regulation increase the levels of active Ras and contribute to the malignant phenotype of cancer cells. Given these activities, it is not surprising that Ras is a target in several human cancers and in a set of genetic diseases termed RASopathies [7–10]. Raf-kinase and PI3K effector pathways are the most well-studied Ras effectors pathways. Given historical challenges in targeting Ras directly, components of these pathways provide a viable alternative. Indeed, there are 28 and 125 studies at different stages of evaluation for RAF/MEK/ERK and PI3K/AKT/mTOR pathways, respectively (http://clinicaltrials.gov). Another important, yet less studied signaling node in the Ras network is the Rac/PAK axis. Sustained Ras signaling requires activation of cytoskeletal modifiers Rac and Rho, typically through the activation of their respective guanine nucleotide exchange factors [11–15]. The Rac/PAK pathway is involved in mechanisms of acquired drug resistance to BRAF monotherapy and BRAF+MEK combination therapy [16, 17]. Thus, targeting a Raf- or PI3K-independent node in the Ras signaling network may provide another therapeutic option.

We targeted the GTP-bound form of the G60A point mutant we previously described [18]. We justify targeting the Ras[G60A] structure by virtual screening as follows. First, a potential binding site cleft is situated between the switch 1 and the triphosphate nucleotide. This cleft results from the restructuring of the switch 1 region in the GTP- but not GDP-bound structure of this mutant. This conformation is different from wild-type Ras, but similar to nucleotide-free Ras in complex with the guanine nucleotide exchange factor Sos [19]. We refer to this as 'the open conformation' of Ras. Switch 1 corresponds to residues 24–40 and is responsible for GTP-/$Mg^{2+}$-coordination and binding to effectors and regulators [20–22]. Second, this mutation severely attenuates the binding of Ras to its effector, Raf kinase, *in vitro* [18] and reverts the transforming ability of constitutively active Ras in cells [23–25]. Third, and most importantly, it was previously shown using solution $^{31}$P-NMR spectroscopy that GTP-bound Ras adopts two conformations in equilibrium [26–28]: one is capable of effector binding and therefore signaling, while the other is non-signaling and is mimicked by the G60A and T35S mutants [29–31]. Taken together, we argue that a small molecule that keeps Ras in the open conformation would inhibit its signaling. A similar approach has previously led to the discovery of the anti-Ras 'Kobe' compounds [32].

Combined data from virtual screening, cell growth and colony formation assays, and chemical synthesis and analysis identify a small molecule termed IODVA1, with cellular inhibitory activity against several transformed cell lines including Ras-driven cells. We show that IODVA1 targets Rac activation and signaling instead of Ras. Further, we demonstrate that

IODVA1 has *in vivo* activity against human MDA-MB-231 triple negative breast cancer (TNBC) and H2122 non-small cell lung cancer (NSCLC) cell lines in tumor xenograft models.

## Results

### Docking

Using the program Autodock [33–35], we performed virtual screening for small molecules that could potentially fit into the identified Ras binding interface pocket. The pocket is lined by Ile21, Gln22, switch 1 residues Gln25-Pro34, Lys147 and Arg149 and the GTP-ribose (**Fig 1A**). Approximately 118,500 compounds from the NCI/DTP Open Chemical Repository were used to identify potential binders. We designed a grid box incorporating the Ras pocket for docking of compounds and searched for the ones with lowest possible binding energy. Autodock returned predicted binding poses, which were grouped into binding clusters with a root-mean-square deviation (rmsd) tolerance of less than 1 Å between poses of the cluster. The

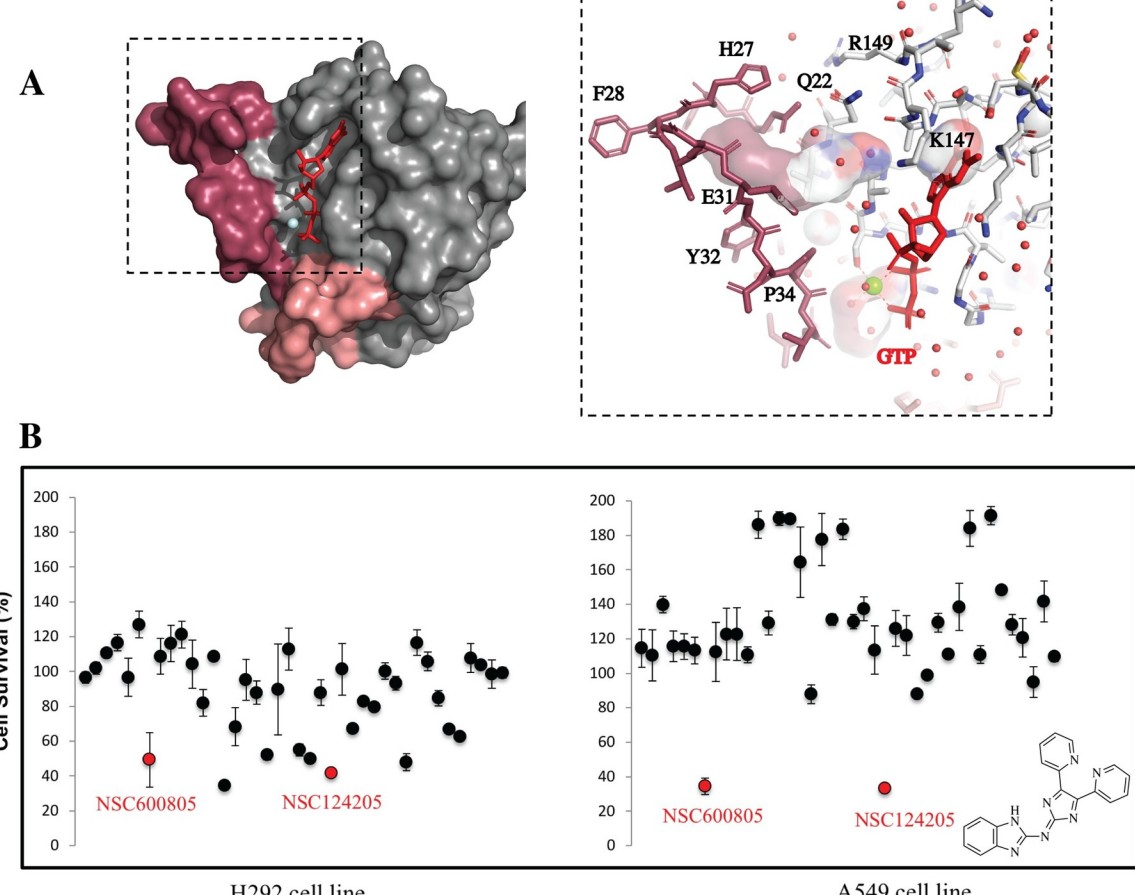

**Fig 1. Identification of NSC124205.** (A) Surface representation (*left*) of the Ras$^{G60A}$ in the GTP-bound form (PDB ID 1XCM) with the nucleotide in red ball-and-stick representation and the Mg$^{2+}$-ion as a cyan ball. Switch 1 region is in maroon and switch II in salmon pink. Zoom in on the pocket used in the docking experiments (*right*). Water molecules are shown as red balls. The cavity between switch 1 and GTP used for *in silico* docking is shown. Figure prepared with the program Pymol. (B) NSC124205 inhibits H292 and A549 cell proliferation. H292 and A549 cells were plated at 500 cells in 96-well plates in 3 technical replicates and treated with vehicle control or each of the 40 top scoring NCI compounds at 10 μM. Cell proliferation was determined by the MTS assay and plotted at the 4-day time point relative to vehicle control. Results are shown as mean ± stdev and are representative of 2 independent experiments. Structure of NSC124205 as shown on PubChem, drawn with ChemDraw 18.0.

results were evaluated by ranking various poses toward the predicted binding energy. Cluster analysis was subsequently accomplished on the basis of rmsd values with respect to the starting ligand geometry. Docked conformations with the most favorable score (predicted binding free energy) and from more populated clusters were selected as the best results. For each cluster, the estimated free energy of binding in kcal/mol was obtained and an estimated inhibition constant ($K_i$) at 298.15 K was derived. Forty compounds with the highest $K_i$ were obtained from the NCI Developmental Therapeutics Program (NCI/DTP, http://dtp.nci.nih.gov). The compounds were dissolved in DMSO as 10 mM stock solutions or lower when poorly soluble.

## Cellular and biochemical phenotypic screening assays

We evaluated the 40 compounds based on the following phenotypic criteria: inhibition of cell proliferation, downregulation of key growth pathways, and inhibition of anchorage-independent growth.

**Proliferation.** To assess the effect on proliferation, compounds were tested in cellular assays at 10 μM against the human lung mucoepidermoid carcinoma cell line H292 and the human lung adenocarcinoma cell line A549 using the MTS cell proliferation assay [36–38]. We chose these two epithelial cell lines because H292 encodes wild-type RAS (RAS$^{WT}$) while A549 encodes KRAS$^{G12V}$ mutation with the expectation that compounds that differentiate between these two cell lines should be specific for oncogenic *vs*. RAS$^{WT}$. **Fig 1B** shows that except for NSC600805 and NSC124205, which significantly decreased proliferation of A549 cells compared to vehicle control, most compounds had little to no effect on this cell line (mean relative survival = 127.7%, STDEV = 36.0%). The averaged percent survival of A549 cells following 4-day incubation by NSC600805 and NSC124205 relative to DMSO vehicle control was 34.4% and 32.9% corresponding to a z-score of 2.60 and 2.63, respectively. When tested on H292 cells, a few compounds from the tested set (mean compounds relative survival = 88.3%, STDEV = 24.1%) decreased its survival by 40% or more but our attempts at identifying one compound that had anti-proliferative effects on A549 but not or less on H292 cells failed. Relative to vehicle control, NSC600805 and NSC124205 percent survival of H292 cells was 49.4% and 41.6% corresponding to a z-score of 1.61 and 1.94, respectively (**Fig 1B**).

Having narrowed the list of compounds to two, we interrogated the NCI Yeast Anticancer Drug Screen database (https://pubchem.ncbi.nlm.nih.gov) about potential anti-cancer activity of both compounds. While NSC124205 at 5 μM inhibited the growth of yeast strains containing single mutations in the *rad50*, *mec2*, and *bub3* genes and double mutant *sgs1* + *mgt1*, *cln2* + *rad14*, and *mlh1* + *rad18* strains at better than 80% compared to untreated controls, NSC600805 showed no inhibitory activity even at 50 μM in any strain. Given the outcome of the phenotypic screen, we thus focused on NSC124205.

**Downregulation of Ras-dependent signaling pathways.** Active Ras binds and activates Raf-kinase and PI3K leading to the activation of ERK and AKT, respectively. Inhibition of Ras signaling is thus most conveniently measured as a decrease in phospho-ERK and -AKT, so we concurrently tested the ability of the compounds to acutely decrease ERK and AKT phosphorylation in 3T3 cells. Cells were serum starved for 24 h, incubated with the DMSO vehicle control or with the compounds at 15 μM for 1 h, EGF stimulated for 5 min, and change in ERK- and AKT-phosphorylation was checked by immunoblotting. NSC124205 caused a significant decrease in both pERK and pAKT compared to candidate compounds **2–8** (**Fig 2A**). To confirm that this decrease was not cell specific, we tested the effect of NSC124205 on the NF1-associated malignant peripheral nerve sheath tumors (MPNST) cell line ST8814, which is characterized by active RAS$^{WT}$ [39, 40]. Following 24 h incubation, NSC124205 at 10 and 50 μM decreased levels of pERK by at least 50%. The decrease in pERK remained unchanged

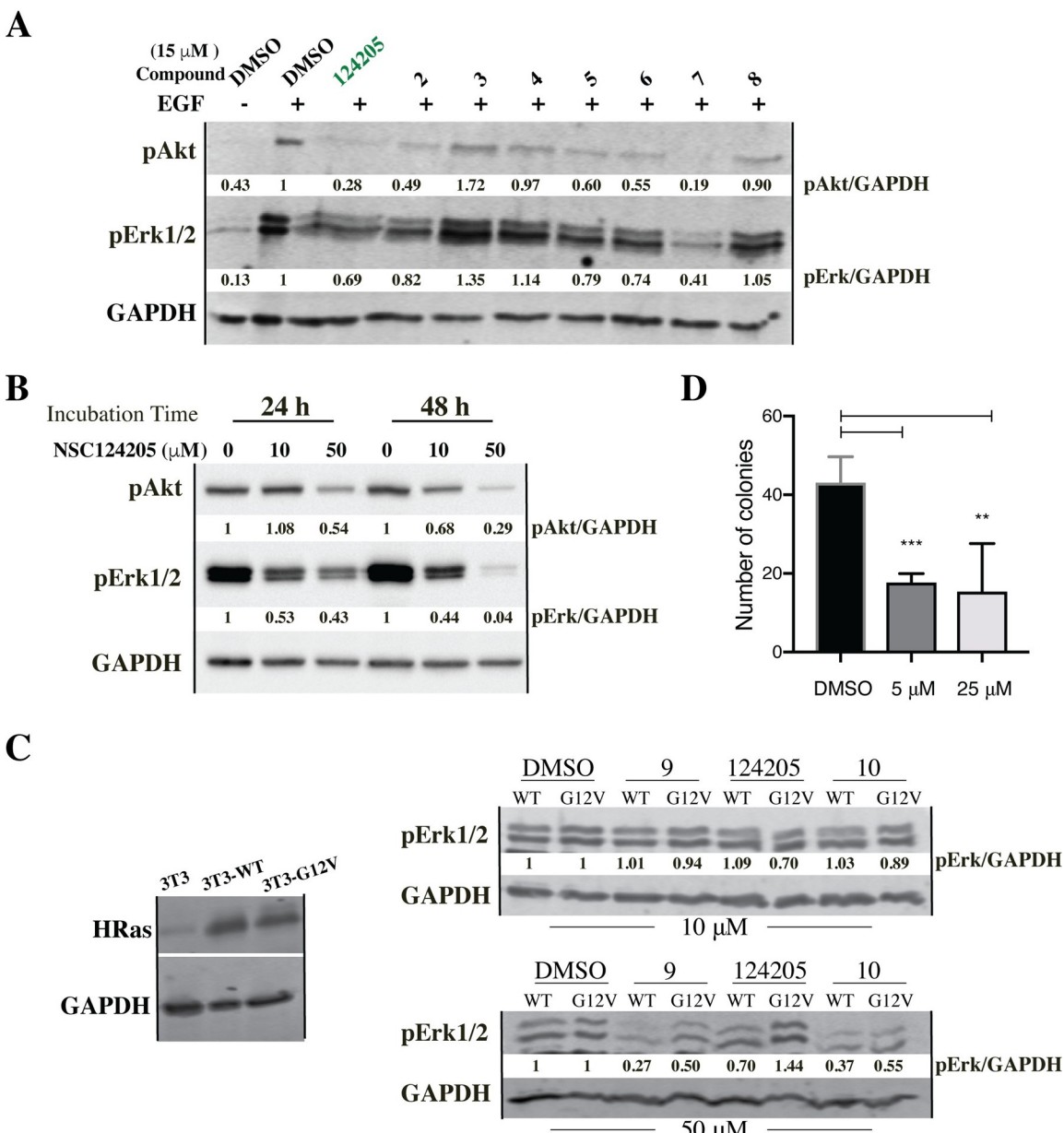

**Fig 2. Biochemical validation of NSC124205.** (A) NSC124205 decreases acute AKT and ERK activation in NIH-3T3 cells. NIH-3T3 cells were starved for 24 h, incubated with DMSO vehicle control or NCI compounds at 15 μM for 1 h, EGF stimulated to 5 min, and lysed with sample buffer. Lysates (30 μg) were resolved on 12% SDS-PAGE, transferred, and blotted for pAKT and pERK1/2. GAPDH was used as loading control and for densitometry analysis of pERK1/2 and pAKT. Results are representative of 2 independent experiments with the order of compound addition into treatment wells and subsequent processing for immunoblotting randomized between two experiments. (B) NSC124205 decreases chronic levels of pAKT and pERK in Ras-activated cells. ST8814 cells were grown in RPMI medium 1640 supplemented with 10% FBS in the presence of indicated concentrations of NSC124205 for 24 or 48 h. Levels of pAKT and pERK were revealed by immunoblotting and quantified in reference to GAPDH levels and normalized to the 0 μM timepoints. Results are representative of 3 independent experiments. (C) NSC124205 has no effect on ERK activation in cells expressing Ras$^{G12V}$. NIH-3T3 cells were stably transduced with full-length HRAS$^{WT}$ or HRAS$^{G12V}$. Cells were lysed in RIPA supplemented with protease and phosphatase inhibitors. Lysates (30 μg) were resolved on 12% SDS-PAGE, transferred, and blotted for HRAS and GAPDH. HRAS$^{WT}$- or HRAS$^{G12V}$- overexpressing 3T3 cells were serum starved for 24 h, incubated with 10 or 50 μM of the listed compounds for 2 h, EGF/serum activated for 5 min, lysed, normalized, separated by the SDS-PAGE, and blotted for pERK. Results are representative of 2 independent experiments. (D) Effects of NSC124205 on colony forming ability of HRAS$^{G12V}$ expressing NIH-3T3 cells. Soft-agar colony formation assays were done with the 3T3 cells expressing HRAS$^{G12V}$ in the presence of NSC124205 at 5 and 25 μM. The data represent means ± stdev of three experiments performed in quadruplicates. The statistical significance of the difference between control and treated cultures was calculated by Student's *t*-test. NCI compounds **2** to **10** are from a different arm of our drug discovery project.

at 48 h for the 10 μM dose and is more pronounced for the 50 μM dose. pAKT level decreased in a time- and dose-dependent manner (Fig 2B). Taken together, our cellular and biochemical data show that NSC124205 decreases proliferation of Ras-driven cell transformation and decreases ERK-phosphorylation albeit at high concentration.

To probe the effect of NSC124205 on oncogenic Ras signaling, we generated NIH-3T3 cells overexpressing wild-type or dominant active HRAS$^{G12V}$ and tested the ability of the compound to interfere with acute ERK activation in these cells. Cells were serum starved for 24 h, incubated with 10 or 50 μM of NSC124205 for 2 h, and EGF/serum activated for 5 min. Cells were lysed, normalized, and cell lysates separated by the SDS-PAGE and blotted for pERK. We did not notice a decrease in pERK in a NSC124205 dose-dependent manner in RAS$^{WT}$ or RAS$^{G12V}$ expressing cells (Fig 2C). Clearly, despite the initial targeted docking, NSC124205 did not inhibit oncogenic Ras.

**Inhibition of anchorage-independent growth.**   Additionally, compounds were screened for their ability to inhibit proliferation of the HRAS$^{G12V}$-overexpressing 3T3 cells using the soft agar colony formation assay at concentrations of 5 and 25 μM. Both NSC124205 concentrations decreased colony formation of the 3T3 cells by 60% with no significant difference between the two concentrations (Fig 2D). Based on its *in vitro* characteristics, we chose NSC124205 for further *in vivo* evaluation.

## Chemical synthesis of IODVA1

Before entering *in vivo* studies, we checked compound NSC124205's identity and purity. High-performance liquid chromatography (HPLC) of an NSC124205 sample showed a mixture of three main constituents which elute at 11.6, 12.8, and 13.4 min in an approximate 2:1:1 ratio, respectively (Fig 3A). The UV profiles of the peaks at 12.8 and 13.4 min showed similar absorbance profiles with maxima close to 250 nm and 300 nm. The mixture was analyzed by High Resolution LCMS using a similar column and conditions. The first peak at 11.6 min (peak **1a**) had a mass-to-charge ratio m/z of 527.21607 [M+H]$^+$ and 264.11178 [M+2H$^+$], corresponding to an elemental composition of C$_{28}$H$_{23}$N$_{12}{}^+$ (theoretical: 527.21632). This peak corresponds to the product of a guanidinobenzimidazole addition to NSC124205. This is in line with the observation that its spectrum lacks red-shifted absorbance maxima due to lack of the extended aromatic system. The other two peaks at 12.8 and 13.4 min have nearly the same m/z 370.14096 [M+H]$^+$ and 370.14097 [M+H]$^+$ respectively, both corresponding to elemental composition, C$_{20}$H$_{16}$N$_7$O$^+$ (theoretical: 370.14108). The minor peaks at m/z 392.12286 and 392.12289 respectively, correspond to [M'+Na$^+$] (S1A Fig). Both peaks possessed similar absorbance spectra suggesting isomers. Neither peak gave a mass corresponding to the structure of NSC124205 as reported by the NCI (Fig 1B), which has a calculated mass of 352.1305 Da. The mass difference of 18 Da suggests the presence of additional O and two H in the structure of NSC124205.

We had accepted the structure shown in Fig 1B as reported on the PubChem and NCI sites, despite our chemistry concerns regarding the central anti-aromatic iminoimidazole ring system. To replicate the probable synthetic attempt toward the NSC124205 structure, we envisioned it as arising from a dual condensation from 2-guanidinobenzimidazole and 2,2'-pyridil or α-pyridoin followed by *in situ* oxidation, analogous to related molecules in the literature [41, 42]. Heating 2-guanidinobenzimidazole and α-pyridoin in dimethyl formamide (DMF) in the presence of acetic acid at 85μC (see Scheme 1) led to a mixture of compounds from which the compound corresponding to peak **1b** was isolated, hereafter referred to as IODVA1. HPLC analysis of the purified IODVA1 shows that a single peak is observed at 12.8 min, which corresponds to peak **1b** observed in the NSC sample (Fig 3A). The UV-profile of the synthesized IODVA1 also matches

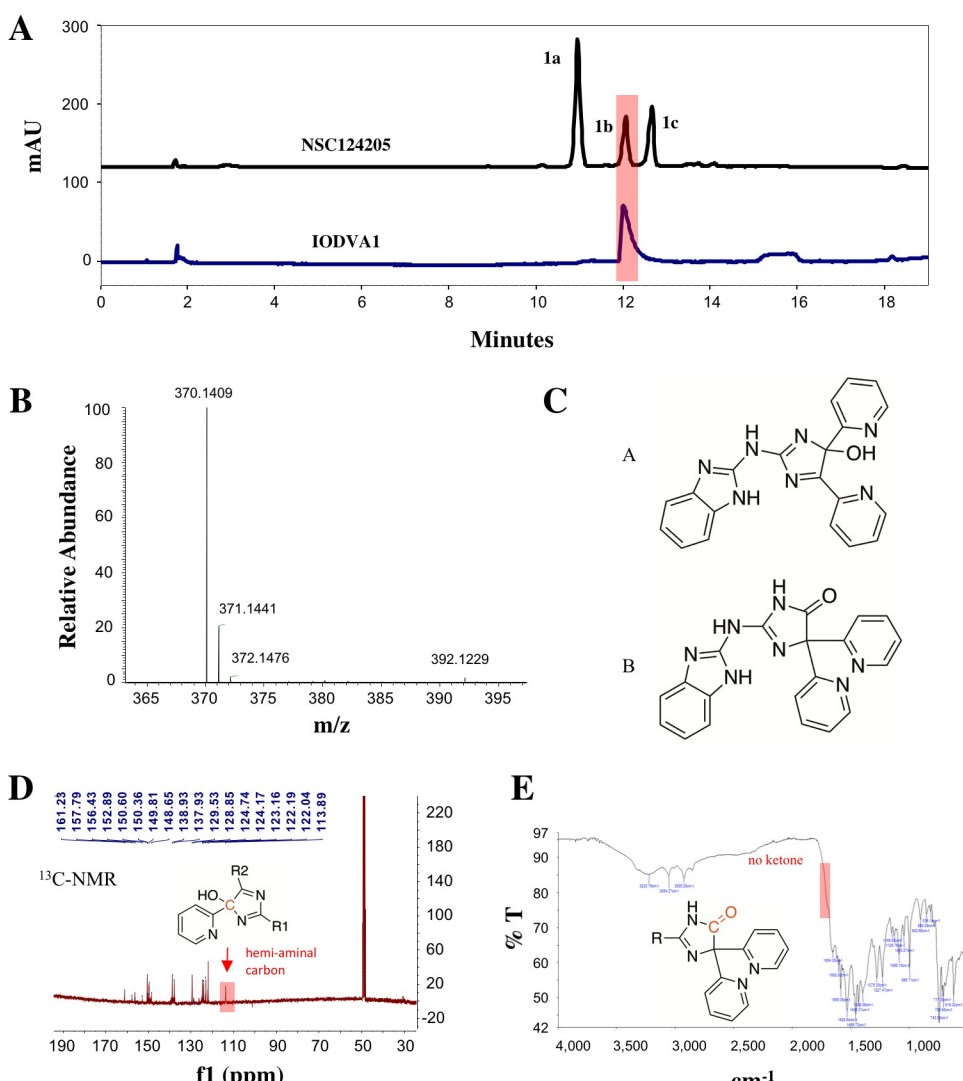

**Fig 3. Chemical analysis of NSC124205.** (A) NSC124205 is a mixture of compounds. 20 μL of a 1 mM NSC124205 solution was loaded on a C18 column washed with buffer A (95% water, 5% acetonitrile, 0.1% formic acid) and a linear gradient was applied over 20 min with buffer B (95% acetonitrile, 5% water, 0.1% formic acid). Three peaks **1a**-**1c** eluted at 11.6 min, 12.8 min, and 13.4 min, respectively (upper chart). Under identical conditions, IODVA1 elutes as a single peak at 12.8 min. (B) Mass spectroscopy electrospray ionization (ESI) spectrum of IODVA1. (C) Proposed structures for IODVA1 with m/z of 370.1409 [M + H]+ following [41] reported reaction of guanidine derivatives with a-diketones. (D) [13]C-NMR of synthesized IODVA1 in methanol-d4. The peak corresponding to the C with the hydroxyl group is boxed. (E) IR-spectrum of IODVA1. The expected stretching in the carbonyl region is boxed.

with peak **1b** of the NCI sample. The MS analysis of the synthesized molecule shows a mass of 370.1409 [M+ H]$^+$ (**Fig 3B**). Nishimura and Kitajima [41] examined the chemistry of guanidines with α-dicarbonyl compounds and found that instead of a dual condensation (*i.e.* formation of two imines from two amines and two carbonyls), one obtained a single condensation to imine and the second condensation stalled at the aminal stage as in structure **A** (**Fig 3C**), not undergoing the second dehydration to the less stable anti-aromatic iminoimidazole system (*i.e.* analogous to NSC124205). Any formation of the NSC124205 structure is likely extremely reactive and either adds a second 2-guanidinobenzimidazole (as seen in the NCI sample peak **1a**) or adds water, reverting to Structure **A**. Structure **A** can also undergo a benzylic acid rearrangement under basic

**Scheme 1. Synthesis of IODVA1.**

conditions to Structure **B** (**Fig 3C**), as described in Luengo [43] and Gupta [44]. Spectral analysis supports the assignment of IODVA1 as Structure **A**. [1]H-NMR analysis (**S2A Fig**) shows 13 protons in the aromatic region with no evidence of symmetry consistent with structure **A** relative to **B**. The [13]C-NMR shows 20 carbon signals and no carbonyl around 170–190 ppm *vs.* a symmetric pattern expected for Structure **B** (**Fig 3D**). Infrared (IR) spectroscopy showed no stretching in the carbonyl region 1680–1720 cm[-1] (**Fig 3E**). MS-MS data shown in **S1B Fig** are also consistent with structure **A**. The spectroscopic data are thus consistent with IODVA1, the synthesized molecule, being Structure **A**. A mechanism describing how 2-guanidinobenzimidazole reacts with α-pyridoin to yield IODVA1 and other byproducts is proposed in **S2B Fig.**

### Effects of IODVA1 on proliferation of cells harboring activated Ras

We tested IODVA1's ability to inhibit the proliferation of cells harboring activated Ras and therefore its ability to recapitulate the sample we obtained from the NCI. We chose the NF1[-/-] RAS[WT] MPNST cell line ST8814 and the triple negative breast cancer cell line MDA-MB-231 carrying the oncogenic KRAS[G13D] mutation [45]. In addition, we used the non-transformed mammary epithelial cell line MCF10A (RAS[WT] immortalized by a spontaneous t(3;9) (3p13;9p22) translocation that deletes the *CDKN2A* gene, also known as p16) [46, 47] and the non-invasive estrogen receptor (ER) and progesterone receptor (PR) positive RAS[WT] breast cancer cell line MCF7 (wild-type p53) and T47D (mutant p53). Cells were grown in complete growth media containing IODVA1 at concentrations between 0.1 and 10 μM or vehicle control and cell number was counted using the trypan blue exclusion method. Increasing concentration of IODVA1 inhibits ST8814 cell proliferation with 50% growth inhibitory concentration (GI50) of 1 μM (**S3 Fig**). Similar results were observed with MCF7, MDA-MB-231, and T47D cells with estimated GI50s ≤ 1 μM (**Fig 4A**). We did not observe appreciable decrease in proliferation in non-transformed MCF10A cells. IODVA1 significantly decreases number of colonies of the breast cancer cells in soft agar at 1 and 3 μM consistent with the cell proliferation results (**Fig 4B**).

 **IODVA1 and Ras activation.** To check if IODVA1 inhibits Ras activation in cells, we determined the levels of active GTP-bound Ras in ST8814 cells treated with IODVA1 (2 μM) or vehicle control at various time points. Glutathione beads-bound GST-RafRBD were incubated with ST8814 cell lysates, thoroughly washed, and the protein complex separated on SDS-PAGE. Levels of GTP-bound pan-Ras proteins bound to RafRBD were determined by immunoblotting. Levels of active Ras between IODVA1 and vehicle control treated cells were similar after 24 and 48 h treatment (**Fig 4C**). However, the decrease in active Ras levels is

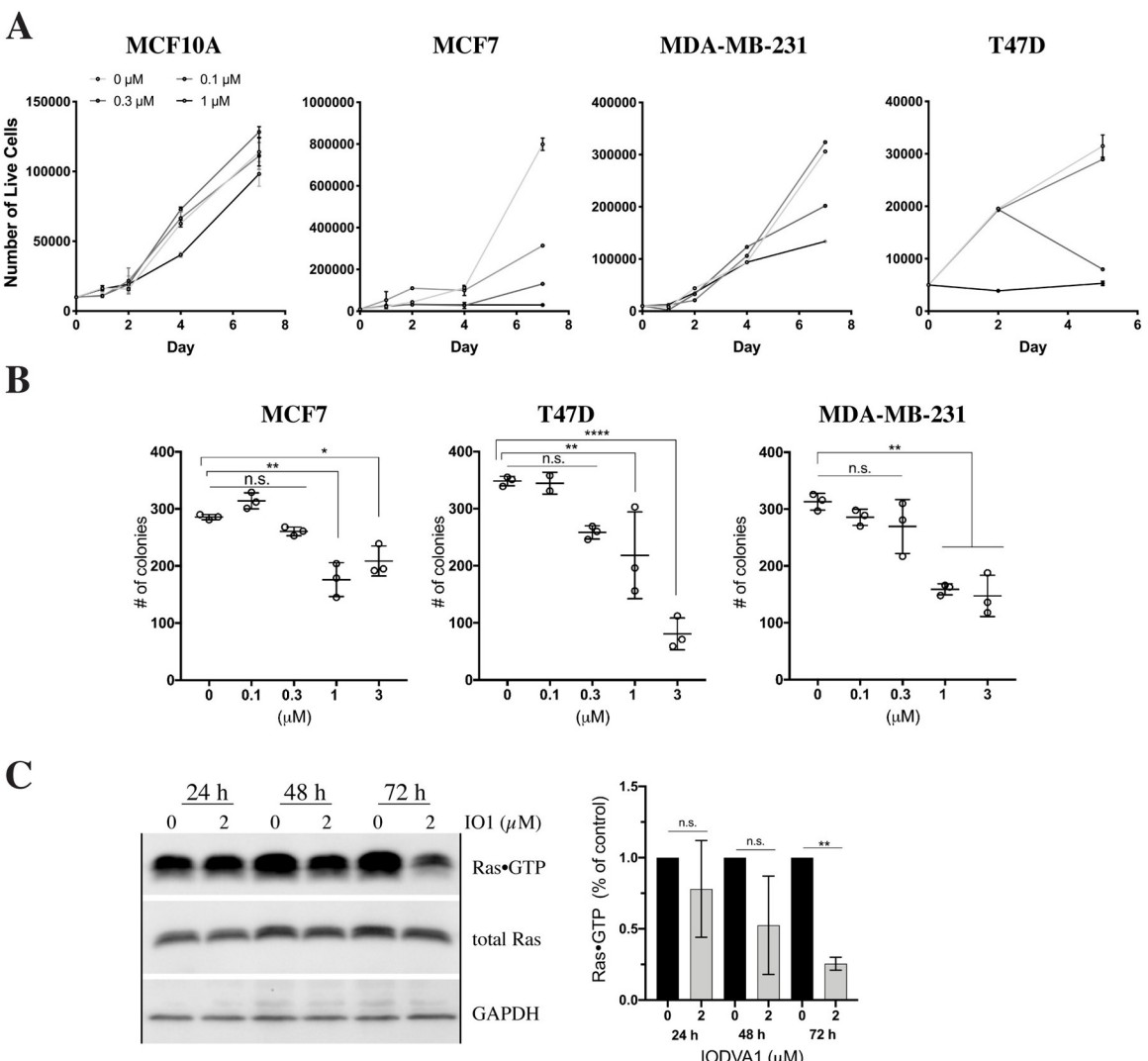

**Fig 4. IODVA1 inhibits proliferation of cancer model cells.** (A) IODVA1 is a potent cell proliferation inhibitor. MCF10A, MCF7, MDA-MB-231, and T47D cells were grown in the presence of the indicated IODVA1 concentrations and counted for up to 7 days. Each dot and bar is the mean ± stdev respectively, of 3 independent experiments, with 2 technical replicates in each experiment. (B) Number of colonies made by MCF7, T47D, and MDA-MB-231 cells at the indicated IODVA1 concentrations. Results shown are mean ± stdev of 2 independent experiments with 4 technical replicates each. (C) IODVA1 deactivates Ras at late incubation times. Total Ras immunoprecipitated with GST-RafRBD from ST8814 cells treated with 0 or 2 μM IODVA1 at the indicated times demonstrate that IODVA1 decreases Ras activation post 48 h treatment. Quantification summary of 2 independent experiments is presented in the graph. n.s.–not significant, *—p < 0.05, **—p < 0.01, ****—p < 0.0001.

noticeable at the 72 h time point. That the levels of active Ras require at least 48 h to decrease suggests that IODVA1 probably does not bind Ras and that its mechanism of action is likely Ras-independent.

## IODVA1 interferes with lamellipodia and circular dorsal ruffle formation and Rac activation

Progression of cancer invasion and metastasis requires the aberrant activation of cell migration, which is driven by the reorganization of the actin cytoskeleton [48–51]. A major trait of Ras-transformed cells is a reorganized actin cytoskeleton, which leads to poor adhesion,

increased motility, invasiveness, and contact-independent growth [5]. To check the effect of IODVA1 on overall actin cytoskeleton structures, we serum starved for 4 h MDA-MB-231 cells, treated with IODVA1 (0–3 μM) for 1 h, then stimulated cells with EGF for 30 min, which induces lamellipodia formation. **Fig 5A** shows that, in comparison to the vehicle-treated cells (0 μM), lamellipodia formation and enrichment of cortical filamentous actin are inhibited with IODVA1 treatment. Cells treated with 1 and 3 μM IODVA1 had more prominent stress fibers and rounded cell shape (open arrow heads). In a parallel experiment, we examined actin cytoskeleton structures of cells stimulated by EGF, followed by wash out and subsequent treatment with IODVA1 or vehicle control. **S4A Fig** shows that overall filamentous actin staining is qualitatively reduced within 30 min of IODVA1 (3 μM) treatment.

Lamellipodium formation and membrane ruffling in response to growth factor stimulation is characteristic of Rac activation [52, 53]. Based on our observation that IODVA1 treatment impedes formation of lamellipodia, we hypothesized that Rac activation may be targeted. We, therefore, evaluated the action of IODVA1 on another Rac-mediated actin structure—circular dorsal ruffles (CDRs) [54]. CDRs are enclosed, dynamic, ring-shaped structures that erect vertically and appear on the dorsal surface of cells and Rac activity is required for their formation [55]. 3T3 fibroblasts were starved for 4 h, treated with IODVA1 (0–3 μM) for 1 h, and stimulated with PDGF for 10 min. **Fig 5B** shows that CDR formation is inhibited with IODVA1 treatment. Cells treated with 1 and 3 μM IODVA1 had intact stress fibers, exhibited "starfish" or triangular cell shape and were devoid of protrusions.

To confirm that IODVA1 interferes with Rac activation, we checked the levels of active Rac and its downstream effector PAK1/2. MDA-MB-231 cells were incubated with IODVA1 (0–3 μM) for 1 h and levels of GTP-bound active Rac were measured by PAK-GBD pull-downs and quantified (**Fig 5C**). IODVA1 significantly decreases levels of active Rac in a dose-dependent manner, levels of the related active Cdc42 GTPase are also decreased but only at the highest IODVA1 concentration. Levels of active RhoA were not affected by IODVA1 (**Fig 5C**) and no effect on stress fiber arrangement was observed (**S4C Fig**). Similarly, MDA-MB-231 cells incubated with IODVA1 (1 μM) for 30 min experience a 50% decrease in levels of pPAK1/2 (T423/T402). Levels of pPAK4/5/6 however, did not change even following 3 h incubation; PAK4/5/6 are primarily Cdc42 specific [56, 57] (**Fig 5D**). Taken together, these data suggest that at low concentrations, IODVA1 inhibits Rac activation and downstream signaling leading to inhibition of lamellipodia and CDR formation.

## IODVA1 may decrease cell-ECM and cell-cell interactions

We then assessed if IODVA1 interferes with cell spreading on the extracellular matrix, an event that is also governed by the rearrangement of the actin cytoskeleton. MCF7, T47D, or MDA-MB-231 cells were plated on fibronectin coated coverslips for 10 min, incubated for additional 30 min in the presence of 0–3 μM IODVA1, fixed and examined by bright field microscopy. **Fig 6A** shows that exposure to IODVA1 interferes with spreading of MCF7 at 0.3 μM and with MDA-MB-231 at 1 μM, as indicated by the decrease in cell area. We did not detect any changes in the area in T47D cells, although we have observed more rounded cells in the presence of IODVA1 (**S4B Fig**). Our results indicate that cells treated with IODVA1 fail to initiate and/or maintain actin reorganization needed for cell spreading and leading-edge formation upon contact with the extracellular matrix and mitogen stimulation, respectively.

*In vitro* 3D assays, such as spheroid formation, serve as an intermediate between 2D (monolayer) cellular assays and *in vivo* animal models. Spheroid formation is mediated by matrix development and remodeling and changes in the cytoskeleton and cell-cell contacts and adhesion [58–61]. To evaluate the effect of IODVA1 on spheroid formation, we plated single cell

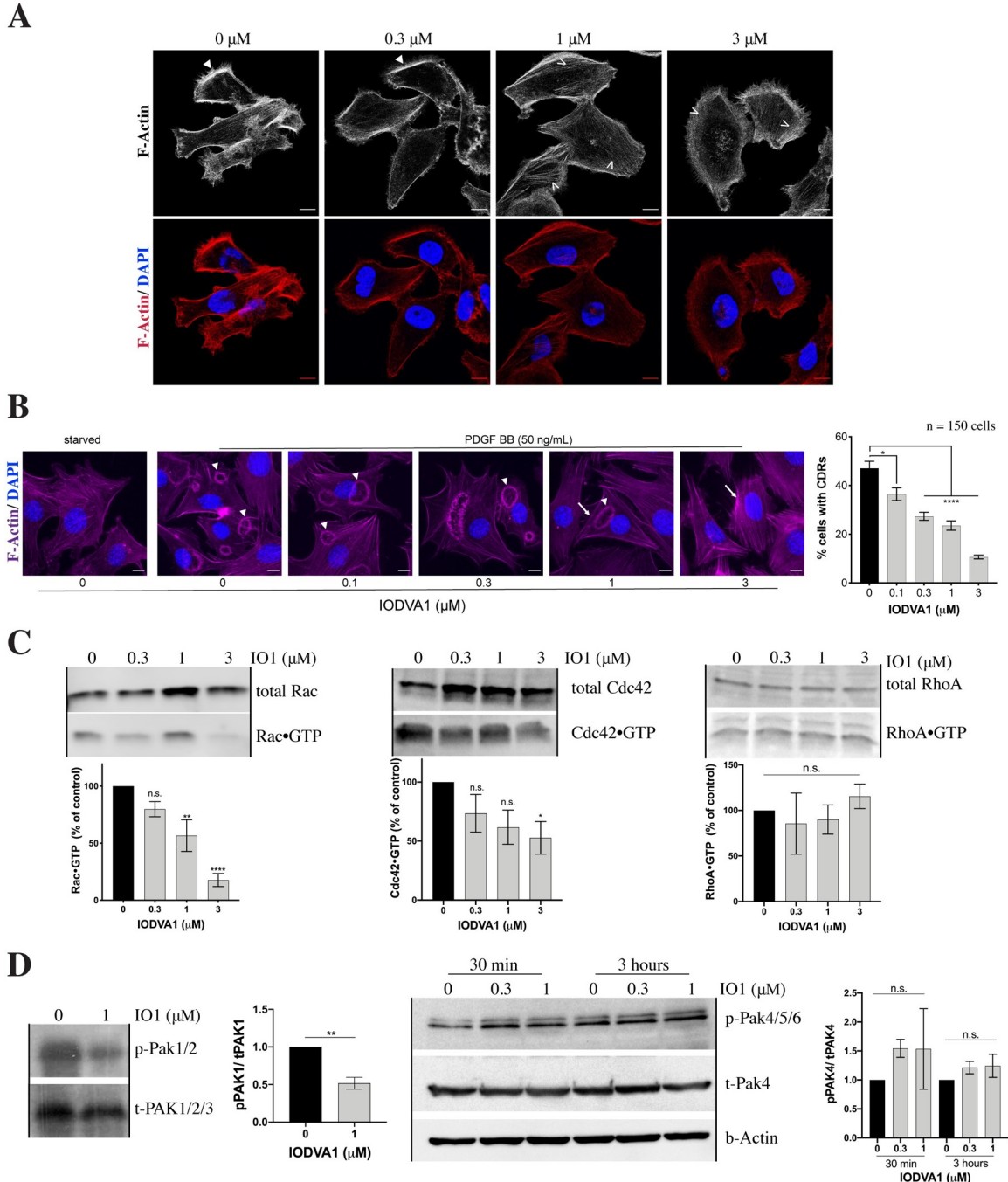

**Fig 5. IODVA1 inhibits lamellipodia and circular dorsal ruffle (CDR) formation in MDA-MB-231 cells and decreases Rac activation.** (A) IODVA1 inhibits EGF-induced lamellipodia formation in MDA-MB-231 cells. MDA-MB-231 cells were plated on fibronectin-coated coverslips, serum starved for 4 h, incubated with the indicated concentrations of IODVA1 for 1 h, then EGF (50 ng/mL) stimulated, fixed and stained with Phalloidin Alexa Fluor 594 (F-Actin, pseudocolored red) and DAPI (nuclei, pseudocolored blue). Representative images show lamellipodia formation and enrichment of actin staining at the leading edge (white closed arrowheads) at 0 and 0.3 μM concentrations and lack of lamellipodia with rounded cell morphology at 1 and 3 μM. Note equally distributed phalloidin staining with presence of stress fibers in the cell body of the 1 and 3 μM treated-cells (white open arrowheads) indicative of stationary cells. Scale bar = 10 μm. Results are representative of three independent experiments. (B) IODVA1 inhibits PDGF-induced CDR formation in 3T3 fibroblasts. NIH-3T3 cells were plated on fibronectin-coated coverslips, serum starved for 4 h, incubated with the indicated concentrations of IODVA1 for 1 h, then PDGF (50 ng/mL) stimulated, fixed and stained with Phalloidin Alexa Fluor 594 (F-Actin, pseudocolored magenta) and DAPI (nuclei, pseudocolored blue). Closed white arrowheads indicate circular dorsal ruffles, arrows indicate lack of elongated morphology typical in stimulated fibroblasts. The percentage of cells with CDRs was counted as the

number of cells with CDRs normalized to the total number of cells in the field. Around 150 cells were counted per condition per experiment. Cells with multiple CDRs were counted only once. Results are representative of three independent experiments. Scale bar = 10 μm. (C) MDA-MB-231 cells were incubated with IODVA1 (IO1) at the indicated concentrations for 1 h, lysed, and incubated with GST-PAK-GBD (binds active Rac and Cdc42) and GST-Rhotekin RBD (binds active RhoA). The protein complexes were resolved on SDS-PAGE and immunoblotted with pan-Rac, Cdc42 or RhoA antibodies. Levels of active Rac (RacGTP, % of control), active Cdc42 (Cdc42GTP, % of control) and active RhoA (RhoAGTP, % of control) were quantified using ImageJ and ImageLab and show combined data as mean ± s.e.m. from at least 3 independent experiments. (D) Left panel—MDA-MB-231 cells were incubated with IODVA1 (IO1, 1 μM) for 30 min, lysed, and immunoblotted for pPAK1(T423)/ pPAK2(T402). Lysates were loaded in two replicates. Results shown are mean ± s.e.m. of two independent experiments. Right panel–MDA-MB-231 cells were incubated with IODVA1 (IO1, 0.3 and 1 μM) for 30 min or 3 hours, lysed and immunoblotted for pPAK4(S474)/ pPAK5(S602)/ PAK6(S560). Results shown are mean ± s.e.m. of two independent experiments. n.s.–not significant, *—p < 0.05, **—p < 0.01, ****—p < 0.0001.

suspension in complete media containing vehicle control or IODVA1 at 0.1–3 μM range using the hanging drop and ultra-low attachment (ULA) methods [62].

In the hanging drop method, spheroids were mixed with the indicated concentrations of IODVA1 in complete media and spheroids were allowed to form in 25 μL hanging drops on the lid of a 10-cm dish. After 96 h, spheroids were transferred into a 10-cm dish using a wide-barrel pipet tip and imaged before and after trituration using bright field microscopy. **Fig 6B** shows that IODVA1-treated MCF7 cells formed smaller spheroids at 1 μM. Mechanical disruption by pipetting resulted in complete dissociation of the spheroids at 1 μM IODVA1. T47D cells treated with 1 μM IODVA1 failed to form packed spheroids, but rather remained as cell aggregates. MDA-MB-231, which form loose aggregates, rather than tight spheroids, were not affected by IODVA1 treatment, but disassembled into smaller aggregates after trituration. No effect was observed in IODVA1-treated MCF10A cells.

For the spheroid formation in the ultra-low attachment plates, MCF7, T47D, and MDA-MB-231 (5,000 cells) were incubated in complete media with IODVA1 at 0–3 μM for 5 days. Aggregates and spheroids were dissociated with Accutase and live cells were counted using trypan blue exclusion. Treatment with IODVA1 significantly decreased the number of live cells in a spheroid (**Fig 6C**). These results indicate that IODVA1 is effective at inhibiting sphere formation in a 3D-culture system probably through inhibition of proliferation.

## IODVA1 does not target kinases

IODVA1 has two pyridine groups attached to a central imidazole group. Pyridine is among the most common scaffolds found in kinase inhibitors [63] and is found in several potent inhibitors of FLT3, Aurora, ROCK, AKT, and other kinases [64–66] suggesting that IODVA1 might have kinase inhibitory activity. To test this hypothesis, we evaluated the potential of IODVA1 to interfere with the ability of 369 recombinant wild-type kinases to hydrolyze ATP. Each kinase was tested twice at one single IODVA1 concentration of 0.5 μM and data were averaged and compared to vehicle DMSO control. A plot of the replicates compared to vehicle control set at 0% is shown in **S5 Fig**. Statistical analysis of the kinome data shows that IODVA1 is ineffective on 98.6% (364 out of 369) of the tested kinases *in vitro*. It shows an inhibition by 22 to 27% of ACK1, TSSK3/STK22C, GSK3b, and IRAK1 and an activation by 21% of YSK4/MAP3K19 at better than 3 standard deviations (> 3σ). However, the inhibition and stimulation effects are modest and higher IODVA1 concentrations are needed to inhibit or stimulate the aforementioned kinases to a level of 50%. Thus, IODVA1 is not a kinase inhibitor and the cellular effects previously observed cannot be explained by kinase inhibition or stimulation given that the cellular GI50s of IODVA1 (0.5 to 1 μM) are similar to the concentration used for the kinase assays.

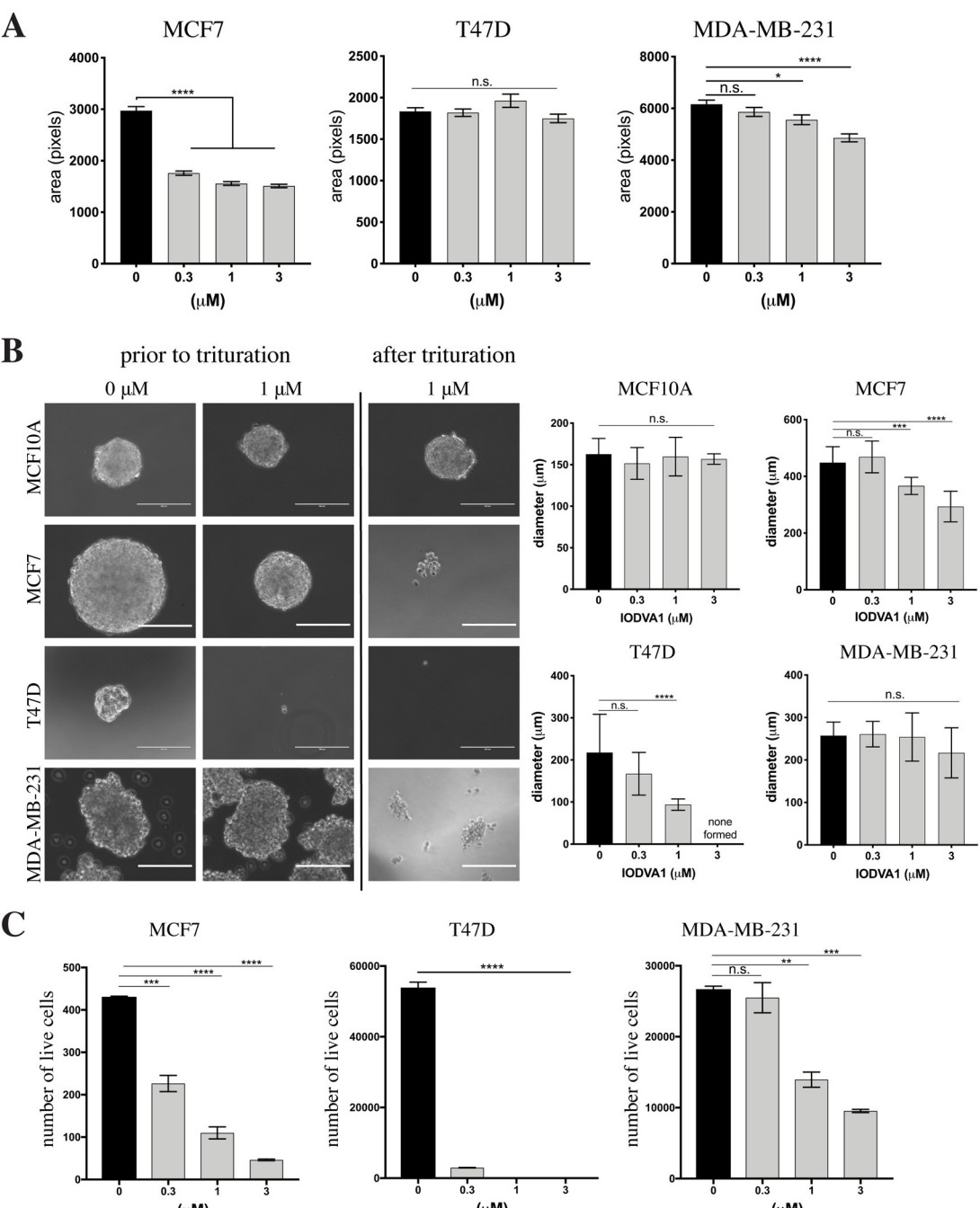

**Fig 6. IODVA1 inhibits cell-substratum and cell-cell interactions.** (A) IODVA1 impedes spreading of MCF7 and MDA-MB-231 cells on fibronectin. MCF7, T47D, and MDA-MB-231 cells were incubated on fibronectin coated coverslips for 10 min, then further incubated with the indicated concentrations of IODVA1 for 30 min in serum-free media, fixed and observed by bright field microscopy. Areas of single cells were calculated from 6 random fields (no less than 300 cells total per treatment group). Results shown are mean ± s.e.m. of a single experiment and are representative of three independent experiments. See also **S4B Fig**. (B) Effects of IODVA1 treatment on spheroid formation in MCF10A, MCF7, T47D, and MDA-MB-231 cells. Left panel—representative bright field images of hanging drop cultures of MCF10A, MCF7, T47D, and MDA-MB-231 cells grown in the absence (0 μM) and presence of IODVA1 (1 μM) prior to and post mechanical pipetting (trituration). Scale bar = 200 μm. Right panel, changes in spheroid/aggregate size due to IODVA1 treatment indicated by the diameter of the spheroids. Results shown are mean ± stdev, N = 15. (C) IODVA1 treatment reduces proliferation capacity in adhesion-free environment. MCF7, T47D, and MDA-MB-231 cells were grown in complete media in the presence of IODVA1 or vehicle control in ultra-low attachment plates for 5 days. Aggregates and spheroids were dissociated with accutase and trituration, and the number of live cells was determined by trypan blue exclusion. Results shown are combined mean ± stdev of two independent experiments. n.s.–not significant, $^*$—p < 0.05, $^{**}$—p < 0.01, $^{***}$—p < 0.001, $^{****}$—p < 0.0001.

## IODVA1 reduces tumor burden of solid tumors *in vivo*

As a proof of concept, we examined if the ability of IODVA1 to reduce oncogene-driven cell proliferation and sphere formation can be translated *in vivo*. We tested its efficacy on one breast cancer and one lung cancer xenograft mouse model. The breast cancer model utilizes the MDA-MB-231 cells. Cells were orthotopically injected into the right and left inguinal mammary fat pads of female nu/nu (nude) mice. Tumor-bearing mice then received an intra-peritoneal (IP) injection of 250 μL of 1 mM IODVA1 every other day, for an average dose of 3.5 mg/kg. Four weeks post treatment a significant decrease of ≥ 50% of tumor volume compared to vehicle-treated control mice was observed (**Fig 7A**). While vehicle-treated tumors doubled in volume, IODVA1-treated tumors failed to grow beyond the pre-treatment tumor size (**Fig 7A**). Tumors were then excised, fixed and paraffin embedded, then stained via immu-nohistochemistry for a proliferation marker (Ki-67), an apoptosis marker (cleaved caspase-3), and with DAPI (**Fig 7B**). Comparison of Ki-67 stained tumors treated with vehicle control and with IODVA1 did not reveal a statistical change in cell proliferation. However, cleaved cas-pase-3 stained tumor sections showed significant increase in apoptosis for cells treated with IODVA1 compared to vehicle control (**Fig 7C**). Thus, IODVA1 has the capacity to induce apoptosis, thereby limiting tumor growth, *in vivo*.

At the end of the four-week treatment when mice were euthanized, peripheral blood was collected via cardiac puncture to check the effect of IODVA1 on white and red blood cells of treated mice for any sign of toxicity. There was no significant difference in total white blood cell counts (WBC), including neutrophils, lymphocytes, and monocytes, nor were there differ-ences in red blood cell counts (RBC), hemoglobin, hematocrit, mean corpuscular volume, mean corpuscular hemoglobin, mean corpuscular hemoglobin volume (MCHV), red cell dis-tribution width (RDW), platelets, and mean platelet volume between blood of drug- and vehi-cle-treated mice (**S6 Fig**). There was also no appreciable difference in animal weights or body condition between control and IODVA1 treated animals.

The lung cancer mouse model was generated by subcutaneous injection of the lung cancer H2122 cells at the right and left flanks of NSG mice. These cells harbor the KRAS$^{G12C}$ mutation and form aggressive tumors. Mice received 7 intraperitoneal (IP) injections of 250 μL of 1 mM IODVA1 or vehicle every other day. Here also, a significant decrease in tumor volume was observed for mice treated with IODVA1 (**Fig 7D**). Tumors were excised and sections were stained with hematoxylin and eosin (H&E) and for Ki67 proliferation marker (**Fig 7E**).

Vehicle-treated H2122 tumors were dense with tumor cells with a high mitotic rate (**Fig 7E,** arrowheads). IODVA1-treated H2122 tumors had a decreased frequency of mitotic cells and increased infiltration of lymphoid and stromal cells, indicative of a therapy-induced immune and fibrotic response. Quantification of the Ki67-positive cells revealed significant decrease in the number of Ki67-positive tumor cells in mice treated with IODVA1 (**Fig 7F**), suggesting that IODVA1 negatively impacts cell proliferation *in vivo*.

Taken together, our *in vivo* data suggest that IODVA1 is efficacious at treating solid tumors, including Ras-driven solid tumors, likely by increasing tumor cell apoptosis and decreasing cell proliferation. In addition, IODVA1 administration does not result in adverse effects on bone marrow function since we did not observe any relevant peripheral blood count changes.

## Discussion

The primary motivation for this work was to exploit a cavity in the crystal structure of the GTP-bound form of Ras$^{G60A}$ [18, 67] and to target it by a small molecule that restraints the switch 1 loop from coordinating the Mg$^{2+}$-ion and adopting a signaling conformation. Com-bining *in silico* screening with cell proliferation and colony formation assays, we identified

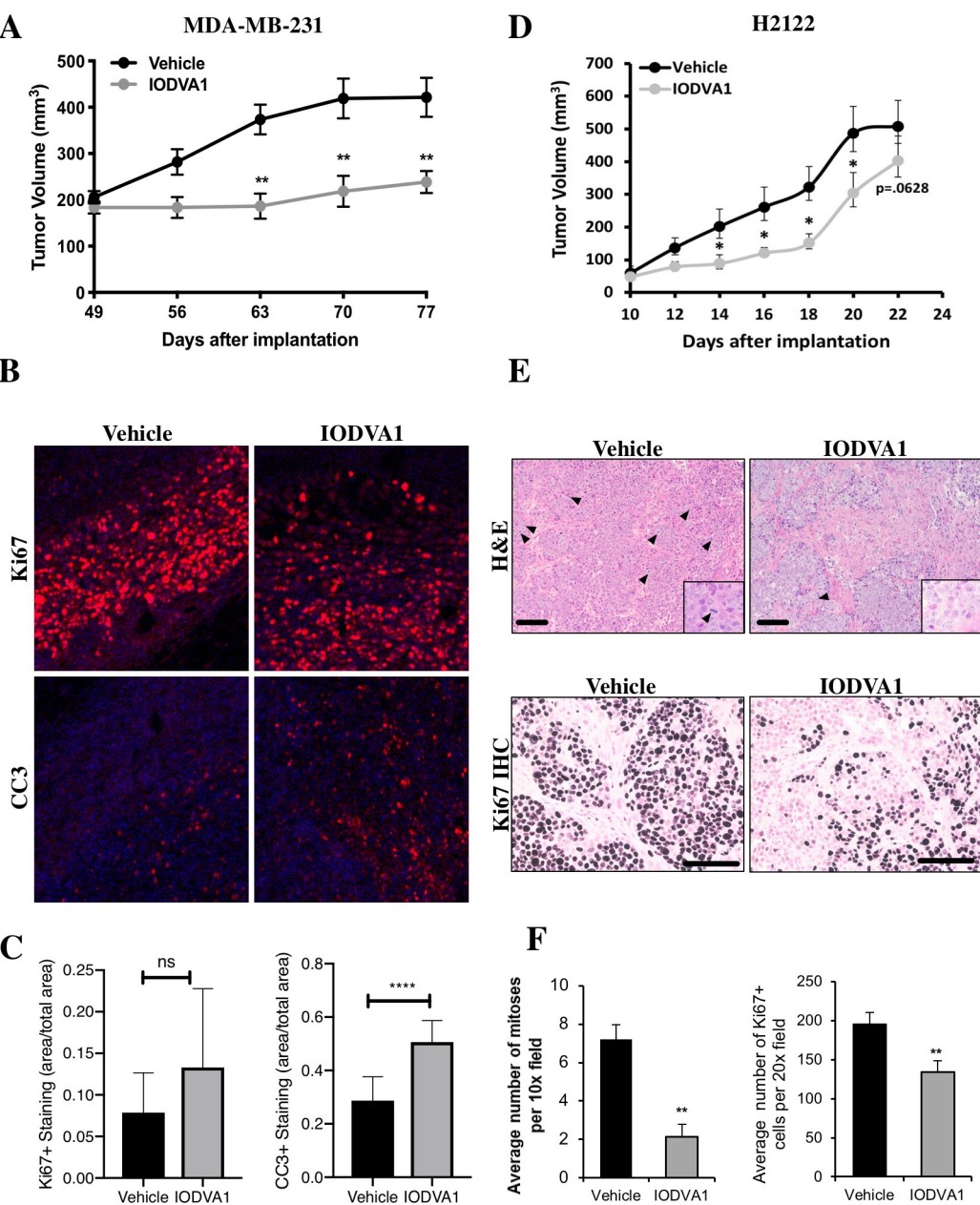

**Fig 7. IODVA1 inhibits tumor growth of human breast and lung cancer xenografts.** (A) Orthotopic xenografts of MDA-MB-231 triple negative breast cancer cells demonstrated decreased tumor growth with IODVA1 treatment. Animals began treatment with vehicle (N = 6) or IODVA1 (N = 5) when tumors reached 200 mm³ in volume (day 49 post-injection) and received IODVA1 treatment three times per week for the next 28 days. (B) Tumors were also stained for Ki67 as a proliferation marker (top panel). IODVA1 treated tumors had a higher percentage of apoptotic cells compared to vehicle treated tumors as detected by cleaved caspase 3 immunofluorescence (CC3, lower panel). (C) Quantification of Ki67-positive and cleaved caspase 3-positive cells of tumors (N = 4) shown in (B). (D) Xenograft tumors of H2122 lung cancer cells demonstrated decreased tumor growth with IODVA1 treatment. Animals began treatment when tumors became detectable at 10 days post-injection (*, p < 0.05). (E) H&E staining of representative H2122 tumors indicates that IODVA1 decreases the number of mitotic cells in the tumor (top, arrowheads). IODVA1 treated tumors also had fewer proliferating cells compared to control vehicle treated animals, as determined by immunohistochemical staining for Ki67 (bottom) and increased intratumoral fibrosis. Representative images were taken at 100x magnification, scale bar is 200 μm. (F) Quantification of mitotic count and Ki67+ cells in control- and IODVA1-treated tumors shown in (E).

NSC124205, an inhibitor of the proliferation of transformed cells at low micromolar concentration with inhibitory effects on proteins downstream of activated Ras signaling. However, HPLC/MS analysis showed that NSC124205 is a mixture of at least three components. Other investigators have reported incorrect annotation of chemical structures in high throughput screen libraries [68, 69] and a high percentage of commercially available chemicals that failed quality control by UPLC [70]. Given a positive result in the phenotypic screen, this prompted us to identify the active ingredient in the NSC124205 mixture. Following a simple synthesis scheme in an attempt to resynthesize NSC124205, we synthesized a small molecule with drug-like properties we termed IODVA1. HPLC and MS analysis identified the chemicals in the three peaks with IODVA1 being peak **1b** of the NSC mixture (Figs **3A and S1A**). We argue that IODVA1 is one active ingredient in NSC124205.

The fact that IODVA1 has cell anti-proliferative activity and halts the growth of an oncogenic Ras-driven lung tumor and of a TNBC mouse xenograft at an unoptimized dose with no apparent toxicity makes it a valuable tool for *in vivo* cancer studies. Further work is required to determine the maximum tolerated dose and the optimized dose for therapeutic efficacy *in vivo*. While its exact mechanism of action is currently being investigated in our laboratory, our biochemical and cellular data suggest that it is not directly targeting Ras. Consistent with this idea is the deregulation of actin structures within minutes of cell exposure to IODVA1 while longer time is needed to deactivate Ras. Our *in vitro* kinome data suggest that it is not a kinase inhibitor either.

The high sensitivity of RAS$^{WT}$ expressing MCF7 and T47D cells and oncogenic Ras expressing MDA-MB-231 to IODVA1 suggest that IODVA1 targets a node used by Ras to regulate cell survival and cell spreading and movement including lamellipodia formation. One such node is the small GTPase Rac [11, 71]. The biochemical observations that IODVA1 decreases Rac activation and signaling, *e.g.* PAK1 activity (**Fig 5C and 5D**) suggest that IODVA1 targets Rac signaling. Our cellular and *in vivo* data support this idea. For example, IODVA1's ability to inhibit formation of Rac-driven cellular actin suprastructures, including lamellipodia and CDRs (**Fig 5A and 5B**) and affect cell spreading and shape (**Fig 6A**) within minutes of cell exposure, as well as cell-cell adhesion (**Fig 6B and 6C**), is consistent with IODVA1 targeting Rac activity. The role of Rac in regulating the organization of the actin cytoskeleton has long been recognized [72–74]. The increase in *in vivo* cleaved caspase-3 and decrease in proliferation in the MDA-MB-231 and H2122 xenograft tumors, respectively indicate that IODVA1 targets a node required by Ras to regulates cell survival and proliferation such as Rac. The role of Rac in regulating Ras-driven tumorigenesis *in vivo* has been reported in various mouse models [71, 75–79]. IODVA1 seems to be specific to Rac as it affects Cdc42 but only at high concentrations and has no effect on Rho. This specificity is consistent with the inability of IODVA1 to decrease levels of pPAK4/5/6 downstream of Cdc42 or remodel stress fibers downstream of Rho. It is plausible that IODVA1 binds and inhibits Rac directly or instead one of its protein regulators. We are testing such scenarios.

IODVA1 is a 2-guanidinobenzimidazole derivative. 2-guanidinobenzimidazole has been shown to block the open conformation of the voltage-gated proton channel Hv1 with an IC50 of 38 μM [80, 81]. In addition, benzimidazole derivatives exhibit a broad spectrum of pharmacological activities such as analgesic, anti-inflammatory, antihistaminic, proton pump inhibitors, uricosuric, cardiotonic, anti-parasitic, anti-viral, and anti-cancer activities [82, 83]. IODVA1 could thus be targeting the proton channel Hv1 or any of the aforementioned benzimidazole derivatives' targets responsible for their broad activities. However, this is unlikely given our cellular and *in vivo* data.

In conclusion, we have identified a di-pyridine guanidinobenzimidazole derivative with *in vitro* and *in vivo* activity against cell lines xenograft models of cancer. Our small molecule

significantly reduces the proliferation of a variety of cancer model cells with diverse genetic lesions and it does so at a low micromolar concentration. The *in vivo* data show that it inhibits tumor growth by increasing apoptosis and/or decreasing proliferation of cancer cells. Our cellular studies and studies of peripheral blood from mice treated with IODVA1 for four weeks (12 doses) that revealed no apparent toxicity suggest that IODVA1 may be specific to transformed cells. Our preliminary cellular and *in vivo* data are consistent with IODVA1 targeting Rac signaling. Experiments to identify the mechanism of action of IODVA1 using click chemistry and other targeted approaches are underway.

## Significance

Small GTP-binding proteins of the Ras superfamily regulate key cellular processes and are involved in many aspects of cell transformation. They also regulate resistance mechanisms to cancer chemo- and targeted-therapies used in the clinic. Finding small molecules that decrease the activity of oncogenic and hyperactivated Ras-like proteins and that can be translated into the clinic has been challenging. And although availability of chemical libraries increased potential of finding an active small molecule, compound mischaracterization may hinder drug discovery efforts. We screened small molecules selected by an *in silico* approach to inhibit Ras from adopting an active conformation in a 2D proliferation and a 3D sphere formation assays against Ras-transformed cells. One NCI compound was selected for its ability to fulfill all three criteria. Chemical analysis showed that the NCI compound was a mixture of at least three components. IODVA1 was synthesized and identified as the active component. Biochemical, cellular, and *in vivo* assays show that IODVA1 is potent on a variety of transformed cells including Ras-transformed cells likely by decreasing Rac signaling. This study thus identifies IODVA1 as a small molecule tool that holds promise for future therapeutic development.

## Materials and methods

### Computational virtual screening

Virtual Screening was performed to identify candidate molecules that would stabilize the open conformation of Ras by targeting the cleft situated between the switch 1 and the triphosphate nucleotide in the crystal structure of HRas[G60A] (PDB ID: 1XCM) (**Fig 1A**) for which position 60 was restituted to Gly. The docking simulations for the virtual screening were performed using rigid body docking, as implemented in AutoDock ver. 4.2 [35], in conjunction with the Cincinnati Children's Hospital Medical Center (CCHMC) Protein Informatics Core' computational pipelines on a Linux cluster with upwards of 512 CPUs. Polyview-3D (http://polyview.cchmc.org) was used to analyze the protein structures and guide the choice of simulation boxes.

A subset of 118,500 drug-like synthetic compounds from the NCI/DTP Open Chemical Repository (http://dtp.cancer.gov) was used for virtual screening. These compounds were derived from the NCI Plated 2007 deposited in the Zinc library (http://zinc.docking.org/catalogs/ncip) by using chemoinformatic filters as described in [84, 85] (OpenEye Scientific Software). 3D-structures for the resulting subset of 118,500 compounds were downloaded from ZINC. Gesteiger partial charges were used for both Ras and chemical compounds. Screening was performed in three stages, using increasingly stringent parameters and gradually more extensive sampling. The latter was achieved by increasing the number of energy evaluations (from $2 \times 10^5$ to $1 \times 10^7$), Genetic Algorithm runs (from 20 to 50) and population size (from 75 to 150), as previously discussed [86]. After initial fast screening, 30,000 top candidates with the highest estimated binding affinities were retained, and subsequently re-scored using improved sampling in the refinement stage. 3,000 top hits were then re-scored using extensive sampling and assessed further to select candidates for experimental validation.

These candidate compounds were ranked based on their estimated binding affinities and entropy of clustering of docking poses in multiple runs of docking simulations resulting in a set of 299 NCI library hits that had entropy of docking poses below 0.2 and predicted median binding constants of less than 10 μM. These top hits were subsequently clustered based on their chemical similarity using Chemmine (http://chemmine.ucr.edu) to further select candidates for experimental validation, while avoiding over-representation of some classes of chemicals, and to visually analyze candidate compounds.

From this joint set, a subset of 40 compounds representing different clusters of chemicals were selected for experimental screening based upon assessment of drug-like properties, similarity to classes of compounds often identified in virtual screening as false positives, and availability of compounds from the NCI/DTP Open Chemical Repository (http://dtp.cancer.gov).

## Plasmids, cell lines, and reagents

MDA-MB-231, MCF7, T47D, MCF10A and HEK239T cells were obtained from ATCC and have since been verified by DNA Diagnostics Center (Fairfield, OH) during the course of these studies. NIH-3T3 fibroblasts were a kind gift from Dr. Susanne I. Wells, ST8814 cells were a kind gift from Dr. Nancy Ratner, A549 and H292 cells were a kind gift from Dr. Jeffrey Whitsett. MDA-MB-231 cells were maintained in Improved MEM media (Invitrogen) supplemented with 10% FBS, 1% penicillin/streptomycin, and 1% amphotericin B. MCF10A were maintained in DMEM/F12 (Invitrogen) supplemented with 5% horse serum, 20 ng/mL EGF, 0.5 mg/mL hydrocortisone, 100 ng/mL cholera toxin and 10 μg/mL insulin. MCF7 and T47D were maintained in DMEM, supplemented with 10% FBS and 10 μg/mL insulin. HEK293T and NIH/3T3 cells were maintained with DMEM supplemented with 10% FBS and 1% penicillin/streptomycin. ST8814, A549, and H292 cells were grown in RPMI supplemented with 10% FBS. Cells were treated with the indicated concentrations of compounds. Control cells were treated with equal volumes of diluent only. The following antibodies were used–GAPDH (GeneTex GTX627408, 1:5000), b-Actin (CST 4970S, 1:10,000), pERK1/2 T202/Y204 (CST 4370, 1:2000), total ERK1/2 (CST 4696, 1:2000), pAKT S473 (CST 9271, 1:1000), pPAK1 T423/ pPAK2 T402 (CST 2601S, 1:1000), PAK1/2/3 (CST 2604, 1:2000), PAK1 (CST 2602, 1:1000), pPAK4 S474/pPAK5 S602/ pPAK6 S560 (CST 3241, 1:1000), PAK4 (CST 62690, 1:1000), anti-mouse Eu (Molecular Devices R8205), anti-rabbit Eu (Molecular Devices R8204), anti-mouse-HRP (CST 7076, 1:5000), and anti-rabbit-HRP (CST 7074, 1:5000), Ki67 (Abcam, IHC 1:100), cleaved caspase 3 (Asp175, CST 9661, IHC 1:100), goat anti-rabbit Alexa Fluor 568 (Abcam, IF 1:500). Fluorescent phalloidins and DAPI were from Invitrogen.

## MTS assays

The colorimetric CellTiter 96 Aqueous One Solution Cell Proliferation Assay (MTS, Promega) was used to determine the number of viable cells and evaluate effect of compounds on cell proliferation. Measurements were made as per supplier's protocol. Assays were performed by adding 10 μL of MTS reagent directly to the culture wells followed by 1 h incubation at 37˚C. The amount of formazan obtained at the end of incubation was measured by absorbance at 490 nm in a 96-well plate reader (Molecular Devices; Sunnyvale, CA). Each 96-well plate had a set of four wells containing medium only and a set of four wells containing cells treated with DMSO vehicle control. Background absorbance was first evaluated from the set of wells containing medium only, averaged, and subtracted from each well. Background corrected absorbance readings were then normalized to and expressed as a relative percentage of the plate averaged DMSO vehicle control. Each experiment was repeated twice per cell line and order of compound arrangement in plates was randomized in different experiments.

## Chemical synthesis

All chemicals, reagents and solvents were purchased from Sigma-Aldrich, Ark Pharm Inc., and Fisher Scientific. Indicated reaction temperatures refer to those of the reaction bath, while room temperature (RT) is noted as 25˚C. Analytical thin layer chromatography (TLC) was performed with glass backed silica plates (20 x 20 cm, pH = 5, MF254). Visualization was accomplished using a 254 nm UV lamp. [1]H- and [13]C-NMR spectra were recorded on a Bruker Avance 400 MHz spectrometer using solutions of samples in methanol-d6. Chemical shifts are reported in ppm with tetramethylsilane as standard. Data are reported as follows: chemical shift, number of protons, multiplicity (s = singlet, d = doublet, dd = doublet of doublet, t = triplet, q = quartet, b = broad, m = multiplet). All compounds were characterized by [1]H-NMR, [13]C-NMR and high resolution mass spectroscopy (HRMS).

(Z)-2-((1H-Benzo[d]imidazol-2-yl)imino)-4,5-di(pyridin-2-yl)-2,5-dihydro-1H-imidazol-5-ol (IODVA1)

2-guanidinobenzimidazole (500 mg, 2.86 mmol) and α-pyridoin (1.22 g, 5.72 mmol) were dissolved in N-N-dimethylformamide (5 mL). After the addition of glacial acetic acid (0.2 ml, 3.4 mmol), the reaction was stirred at 85˚C for 48 h. The reaction was cooled then quenched with water, the pH was neutralized, and the aqueous solution was extracted three times with ethyl acetate. The organic layers were combined and washed with water and brine then dried over $Na_2SO_4$ and filtered using filter paper. Silica (2 g) was added before concentrating the solution under reduced pressure. The solid was loaded onto a column then purified with flash chromatography using 1:10 methanol: methylene chloride gradient. The targeted fractions (Rf = 0.5) were collected and reduced under pressure to yield 111 mg (0.30 mmol, 11%) of the desired product as a pale brown-gold solid.

[1]H NMR (400 MHz, MeOD-d6): δ = 7.07 (m, 3H), 7.33 (m, 2H), 7.51 (m, 2H), 7.63 (m, 1H), 7.85 (m, 1H), 8.05 (m, 1H), 8.19 (m, 1H), 8.33 (m, 1H), 8.53 (d, 1H).

[13]C NMR (400 MHz, MeOD-d6): δ = 113.89, 122.04, 122.19, 123.16, 124.17, 124.74, 128.85, 129.53, 137.93, 138.93, 148.65, 149.81, 150.36, 150.60, 152.89, 156.43, 157.79, 161.23.

HRMS-ESI: $[M+H]^+$ ($C_{20}H_{16}N_7O$): calculated: m/z = 370.1411. Found: m/z = 370.1409

## Immunoblotting

Cells were lysed in RIPA buffer, supplemented with 1% SDS, protease (Sigma P8340) and phosphatase inhibitors (Roche 04906845001). Lysates were separated on 12%, 15% or 4–20% SDS-PAGE and transferred onto PVDF membrane (Bio-Rad TurboBlot). Membranes were blocked in 5% BSA in TBS-T (0.05%) and incubated overnight with primary antibodies. Blots were washed, probed with the appropriate secondary antibodies and processed with ECL (film or Bio-Rad chemiluminescent system) or imaged on the SpectraMax i3 platform with ScanLater module (Molecular Devices).

## Retrovirus production and transduction

pBabe puro $HRas^{G12V}$ plasmid was from Addgene (#9051). $HRas^{WT}$ was made by reverting G12V to G using site-directed mutagenesis. Plasmids were verified by Sanger Sequencing (CCHMC DNA Core). Retroviral supernatants were produced by transfecting HEK293T cells with pBabe-puro plasmids with pCL-Eco in 1:1 ratio using Calcium Phosphate method (Trono Lab). Supernatants were harvested 24 and 48 h post-transfection and filtered. NIH/3T3 cells were transduced overnight in the presence of polybrene (10 μg/mL) and selected with puromycin (3 μg/mL). Retroviral production and manipulation were performed in BSL-2 facilities.

## Anchorage-independent growth assays

Bottom agar layer was prepared by mixing 2X complete DMEM with 1% Noble agar (BD) in a 12-well plate for a final concentration of 0.5% and allowed to solidify. For the top agarose layer for each well, 2,500 cells in 2X compete DMEM were resuspended in 0.6% low-melting agarose (IBI Scientific) and layered over the bottom agar layer. The next day, 100 μL of complete growth media containing test compounds was overlaid the upper layer to prevent desiccation. Media was refreshed twice weekly. After 21 days, colonies were stained with 0.1% p-iodonitro tetrazolium violet (Sigma-Aldrich), imaged using an EVOS microscope (Life Technologies), and counted.

## Spheroid formation assay

Spheroids were formed by the hanging drop method or using ultra-low attachment (ULA) plates (Corning). For the hanging drop assay, cells were trypsinized, resuspended at 25,000 cells/mL in complete media containing vehicle (DMSO), or the indicated concentrations of IODVA1, and plated in 25 μL drops on an inverted lid of a 10-cm dish. The dish was filled with 7 mL PBS, the lid replaced and incubated for 3–5 days. For mechanical testing of spheroid compaction, spheroids were first imaged on EVOS microscope, repeatedly pipetted 7–9 times, then imaged again to assess spheroid disruption. For the ULA-based spheroid formation, 5,000 cells were resuspended in 500 μL of complete media containing vehicle (DMSO) or the indicated concentrations of IODVA1 and plated in 24-well ULA plates (Corning). Spheroid formation was monitored daily and imaged using EVOS microscope (Life Technologies). For assessment of attachment-free proliferation, the contents of the well were transferred into an Eppendorf tube, centrifuged at 100 x g for 5 min, dissociated into a single cell suspension using trituration and Accutase (Invitrogen) treatment at room temperature and counted using trypan blue exclusion.

## Active GTPase binding assays

Levels of active Ras, Rac, Cdc42 and RhoA were determined using the Active Ras Pull-Down and Detection Kit, Active Rac Pull-Down and Detection Kit (Thermo Scientific) and RhoA/Rac1/Cdc4 Pull-down Activation Assay Combo Biochem Kit (Cytoskeleton). Cells were cultured and treated as indicated and lysed in buffer provided by the manufacturer. Clarified whole cell lysates were incubated with recombinant GST-Raf1-RBD (for active Ras) (amino acids 1–149) and glutathione beads (both supplied by the manufacturer), GST-Rhotekin-RBD (for active RhoA), GST-PAK-GBD (for active Rac and Cdc42) for 1 h at 4˚C, washed and the resulting complexes eluted from the resin by boiling in 2X SDS sample buffer. Proteins were resolved by SDS-PAGE, transferred to nitrocellulose and the level of active GTPase relative to input lysate were determined by immunoblotting analysis using the anti-Ras, anti-Rac, anti-Cdc42, or anti-RhoA antibody supplied by the manufacturer.

## Kinase assay

The Reaction Biology (http://www.reactionbiology.com) HotSPot miniaturized radioisotope filter binding assay platform was used to measure the activity of 369 wild-type kinases in presence of a single NSC124205 dose [87]. In brief, for each reaction, kinase and substrate were mixed in a buffer containing 20 mM HEPES (pH 7.5), 10 mM $MgCl_2$, 1 mM EGTA, 0.02% Brij35, 0.02 mg/ml BSA, 0.1 mM $Na_3VO_4$, 2 mM DTT, and 1% DMSO. IODVA1 was then added to a final concentration of 0.5 μM to each reaction mixture. After 20 min incubation at room temperature, reaction was initiated by adding ATP (Sigma-Aldrich) and [γ-$^{33}$P]-ATP

(PerkinElmer, specific activity of 10 Ci/L). Reactions were incubated for 2 h at room temperature and spotted onto P81 ion exchange cellulose chromatography paper (Whatman). Filter paper was washed in 0.75% phosphoric acid to remove excess ATP. The percent remaining kinase activity relative to a vehicle-containing (DMSO) kinase reaction was calculated for each kinase/IODVA1 pair. Each kinase inhibition assay was done in duplicates and averaged. Data were processed and analyzed in Excel.

## Immunofluorescence and microscopy

For assessment of lamellipodia initiation and maturation, assays were performed in two ways. 1) MDA-MB-231 were plated on fibronectin-coated coverslips in serum-free media for 4 h, followed by 1 h incubation with IODVA1 (0–3 μM) in serum-free media. Cells were then EGF stimulated (50 ng/mL) for 30 min to induce lamellipodia formation. Cells were fixed in 4% paraformaldehyde, permeabilized in 0.2% Triton-X 100 and stained with Phalloidin Alexa Fluor 568 or 594 and mounted in ProLong Gold DAPI (Life Technologies). 2) MDA-MB-231 cells were seeded at a density of $2 \times 10^4$ cells per chamber in an 8-chamber glass slide with or without EGF (5 ng/ml) for 10 min. After treatment with 1 μM IODVA1, cells were fixed and processed as in 1). Staining was visualized with a Nikon A1R confocal microscope. 10-random fields were imaged and analyzed for lamellipodia formation. For assessment of circular dorsal ruffle formation, NIH-3T3 cells were processed as in [54]. Briefly, cells ($4 \times 10^4$) were plated on fibronectin-coated coverslips, serum-starved for 4 hours and treated with IODVA1 (0–3 μM) in serum-free media for 1 hour. Ruffling was induced with PDGF BB (50ng/mL, Peprotech) for 10 min. Cells were fixed in 4% paraformaldehyde and processed for immunofluorescence microscopy as above. Around 150 cells were counted per treatment group per experiment. For assessment of stress fibers, cells were plated on fibronectin-coated coverslips for 4 hours in serum-free media, incubated with IODVA1 (0–3 μM) for 1 h, fixed and processed with fluorescent phalloidin as before.

For cell spreading, MDA-MB-231, MCF7 and T47D were seeded on fibronectin-coated coverslips in serum-free media. After 10 min, IODVA1 (0–3 μM) was added in serum-free media. After 30 min (40 min total), cells were fixed in 4% paraformaldehyde and imaged using bright field microscopy (EVOS, Life Technologies). Six random fields were imaged under 20X objective and cell area of individual cells were quantified to assess the degree of spreading (ImageJ). Over 300 cells were counted per treatment group per experiment.

## *In vivo* analysis using MDA-MB-231 xenograft mouse model

For xenograft studies, $1 \times 10^6$ MDA-MB-231 cells were suspended in PBS and injected into each inguinal mammary fat pad of nulliparous, 10-week old female athymic nude nu/nu mice. Tumors were measured weekly with digital calipers and volume was calculated as $[(\pi/6) \times L \times W^2]$ [88, 89]. Treatment began eight weeks post-injection, when tumors reached 200 mm$^3$. Mice received intraperitoneal injections of 250 μL of diluent (5% DMSO in PBS) or 1 mM of compound IODVA1 three times weekly for 4 weeks, for an average drug dose of 3.5 mg/kg. At necropsy, the mice were weighed and the tumors were excised, measured, weighed, fixed in 4% paraformaldehyde, and embedded in paraffin. Peripheral blood was collected by cardiac puncture and analyzed with a Hemavet (Drew Scientific, Miami Lakes, FL, USA) for complete blood counts. Usage and handling of mice were performed with the approval of the Cincinnati Children's Institutional Animal Care and Use Committee. All mice were housed in specific pathogen free housing with *ad libitum* access to food and water.

## Histology

Tissues were fixed in 4% paraformaldehyde then paraffin embedded tissues were cut into 5 μm sections. Tissues were stained with either hematoxylin and eosin (H&E) or by immunofluorescence. Tissue sections were subjected to sodium citrate antigen retrieval, blocked with 10% normal goat serum, and incubated with antibodies. Tissues were counterstained with 4′, 6-diamidino-2-phenylindole (DAPI) and coverslipped with VectaShield HardSet (Vector Labs). Images were acquired by confocal microscopy (Nikon) and immunofluorescence analysis was performed using Image J.

## *In vivo* analysis of H2122 xenograft mouse model

Human lung cancer cell line NCI-H2122 (ATCC) harboring biallelic $KRAS^{G12C}$ mutations were cultured in RPMI-1640 supplemented with 10% FBS. For injections, NCI-H2122 cells were trypsinized and suspended at a concentration of $2 \times 10^7$ cells/mL in 50% Matrigel (Corning). 100 μl of cell suspension was injected subcutaneously into 6-week old NOD.Cg-*Prkdc^scid Il2rg^{tm1Wjl}*/SzJ (NSG) mice. Beginning 10 days post-xenograft, mice received intraperitoneal injections every other day for 14 days (7 injections total). A 200 mM IODVA1 stock solution in DMSO was freshly diluted 1:19 in DMSO then subsequently diluted 1:9 in PBS to achieve a 1 mM final concentration in 10% DMSO. Mice were injected with 250 μl of vehicle (10% DMSO) or IODVA1 (1 mM). Tumor size was measured using digital calipers and tumor volume estimated using the formula (length x width$^2$)/2. For immunostaining, paraffin-embedded sections were stained with Ki-67 antibody (clone SP6, ThermoFisher) at a dilution of 1:1,000. DAB stained slides were counterstained with nuclear-fast red stain. For quantification of Ki-67 positive cells, random fields were imaged at 200x magnification and quantified using Image J software. A minimum of 20 random fields were used for quantification. Scale bar = 100 μm.

## Statistical analysis

Statistical analyses were carried out using GraphPad Prism. Comparison of two groups was done using Student's t-test, and comparison of data sets with more than two groups was carried out using ANOVA with multiple comparisons. Alpha was set to 0.5.

## Supporting information

**S1 Raw images.**
(PDF)

**S1 Fig.** (A) Electrospray ionization spectrum of the 3 peaks at 11.6, 12.8, and 13.4 minutes, respectively. Structure of each component is shown. (B) MS-MS fragmentation of the 370.1409 peak of IODVA1.
(TIF)

**S2 Fig.** (A) 1H NMR of synthesized IODVA1 in methanol-d4. (B) Proposed mechanism of the reaction between 2-guanidinobenzimidazole and α-pyridoin resulting in structures A and B, NSC124205, and other products.
(TIF)

**S3 Fig. IODVA1 inhibits proliferation of ST8814 cells.** ST8814 cells were grown in the presence of the indicated IODVA1 concentrations and counted daily for 4 days. Each dot and bar is the mean and standard deviation respectively, of 3 independent experiments.
(TIF)

**S4 Fig.** (A) MDA-MB-231 cells were EGF activated for 10 min, washed, treated with DMSO vehicle control or IODVA1 3 μM for 30 min, fixed, and stained for F-actin and nuclei (N = 3). Arrows point to lamellipodia structures. Images were taken at 100X. Scale bar = 10 μm. (B) Representative images from which main Fig 6A quantification was made. (C) Representative images of NIH-3T3 cells treated with the indicated concentrations of IODVA1 for 1 hours in serum-free media, fixed and stained with fluorescent phalloidin to visualize stress fibers. (TIF)

**S5 Fig. IODVA1 kinome inhibitory activity.** The activity of 369 kinases was tested twice in the presence of 0.5 μM IODVA1. Plotted is the remaining activity of replicate 1 vs 2 expressed as % of vehicle control set at 0% for each kinase. Kinases whose activities were decreased or increased by more than 3σ from mean are indicated. (TIF)

**S6 Fig. Repeated doses of IODVA1 do not cause toxicity in the hematopoietic system.** Peripheral blood collected after 12 doses of IODVA1 in tumor-bearing animals were analyzed for blood counts with a Hemavet. No statistically significant changes in blood counts were detected between vehicle control and IODVA1 treated animals (N = 4, mean ± stdev). (TIF)

## Acknowledgments

We thank Drs. Marie-Dominique Filippi for critical reading of the manuscript, Daniel Starczynowski, Yi Zheng, and their lab members for valuable advice, Nancy Ratner, Susanne Wells, and Jeffrey A. Whitsett for sharing reagents.

## Author Contributions

**Conceptualization:** Nicolas N. Nassar.

**Data curation:** Anjelika Gasilina, Gurdat Premnauth, Purujit Gurjar, Jacek Biesiada, Shailaja Hegde, David Milewski, Gang Ma, Edward Merino, Jarek Meller, William Seibel, Lisa Privette Vinnedge, Nicolas N. Nassar.

**Formal analysis:** Anjelika Gasilina, Edward Merino, Jarek Meller, William Seibel, Lisa Privette Vinnedge, Nicolas N. Nassar.

**Funding acquisition:** Nicolas N. Nassar.

**Investigation:** Tanya V. Kalin, Nicolas N. Nassar.

**Methodology:** Nicolas N. Nassar.

**Project administration:** Nicolas N. Nassar.

**Resources:** Nicolas N. Nassar.

**Supervision:** José A. Cancelas, Nicolas N. Nassar.

**Validation:** Nicolas N. Nassar.

**Visualization:** Anjelika Gasilina, Nicolas N. Nassar.

**Writing – original draft:** Anjelika Gasilina, Gurdat Premnauth, David Milewski, Lisa Privette Vinnedge, Nicolas N. Nassar.

**Writing – review & editing:** Anjelika Gasilina, Nicolas N. Nassar.

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
