## [Decision Letter · Decision Letter 0]

19 Dec 2019

PONE-D-19-30856

IODVA1, a guanidinobenzimidazole derivative, targets Rac activity and Ras-driven cancer models.

PLOS ONE

Dear Dr. Nassar,

Thank you for submitting your manuscript to PLOS ONE. After careful consideration, we feel that it has merit but does not fully meet PLOS ONE’s publication criteria as it currently stands. Therefore, we invite you to submit a revised version of the manuscript that addresses the points raised during the review process.

Additional experiments and controls are needed to confirm the authors' conclusions.  Furthermore, data interpretation should be re-worked according to the results presented and updated.

We would appreciate receiving your revised manuscript by Feb 02 2020 11:59PM. To enhance the reproducibility of your results, we recommend that if applicable you deposit your laboratory protocols in protocols.io, where a protocol can be assigned its own identifier (DOI) such that it can be cited independently in the future. For instructions see: http://journals.plos.org/plosone/s/submission-guidelines#loc-laboratory-protocols

We look forward to receiving your revised manuscript.

Kind regards,

Irina V. Lebedeva, Ph.D.

Academic Editor

PLOS ONE

Journal Requirements:

4. Please upload a new copy of Figure 2, 3, 4 and 6 as the details are not clear. Please follow the link for more information: http://blogs.PLOS.org/everyone/2011/05/10/how-to-check-your-manuscript-image-quality-in-editorial-manager/

Reviewers' comments:

Reviewer's Responses to Questions

**Comments to the Author**

1. Is the manuscript technically sound, and do the data support the conclusions?

Reviewer #1: Partly

Reviewer #2: Yes

2. Has the statistical analysis been performed appropriately and rigorously? 

Reviewer #1: Yes

Reviewer #2: No

3. Have the authors made all data underlying the findings in their manuscript fully available?

Reviewer #1: Yes

Reviewer #2: Yes

4. Is the manuscript presented in an intelligible fashion and written in standard English?

Reviewer #1: Yes

Reviewer #2: Yes

5. Review Comments to the Author

Reviewer #1: In this paper the authors identify IODVA1, a potent small molecule that was found to be active in xenograft mouse models of Ras-driven lung and breast cancers. Insilico screening was performed and that led the authors to bring forward IODVA1. The team used 2 D and 3D culture system, xenograft studies and tumor analysis to show potency of the compound. Mechanistically, the authors demonstrate that IODVA1 can inhibit RhoGTPase Rac and downstream signaling. The approach presented here is distinct from other RAS targeted agents as the authors are targeting GTP-bound form of the G60A point mutant.

Specific comments:

Isogenic cell lines with different ras mutations would be better model to prove the specificity

Not clear how specificity will be achieved in cancer cells. RAS active conformation will also be needed for wt-KRAS function

The animal studies are not very convincing and the tumors seem to rebound to a size that is equivalent to control arm

Rac pathway and downstream effector PAK1 cannot be studied in isolation. Other Group I and Group II PAKs need to be studied

Besides RAC other RhoGTPases and their effectors need to be studied as well

PAK4 and its role in lamellipodia formation is known and the authors should evaluate the additional paks

Background and introduction as well as discussion should include references on the different studies that have looked at RAC and PAK downstream of RAS post SMI treatment

Typos should be checked and corrected

Some of the figures need improvement in the style of presentation

Reviewer #2: This is a very interesting and well-written paper that describes the identification of a novel guanidinobenzimidazole derivative that has significant anti-proliferative effects, as well as effects on cell spreading and actin cytoskeleton reorganization. The way the paper is presented is really interesting, since the authors described in a very clear and sequential manner how the search for an anti-RAS compound ultimately leads to the identification of a compound that targets a RAS effector rather than RAS itself. Moreover, the authors had to overcome an unexpected problem regarding the purity of the compound provided by NCI, but they took this into their advantage and identified the actual compound responsible for the biological activity (IODVA1). The quality of the data is outstanding, and the road towards the identification of the responsible compound and its biological activity is described in such a clear manner that it is a pleasure to read. The authors also were very careful in ruling out effects on kinases, and used multiple cell lines with different genetic backgrounds to achieve their conclusions. Discussion is also very focused.

Of course the main question that remains is how IODVA1 acts at a mechanistic level. This may be an effect on Rac or on Rac-GEF/Rac interactions, as described for other Rac inhibitors, or through other mechanisms. I wish this is part of the current study, but this reviewer fully understands that the authors are currently assessing different scenarios and that it would be too much to add new data to the present manuscript, which is already long and comprehensive in nature. I presume that modeling studies on Rac-GEF/Rac interactions would provide the answer, and certainly look forward to the next study.

Minor comments:

1. The authors should provide a better quantitative analysis of data, specifically densitometry analysis of Western blots and indication of reproducibility/mean+/-S.E.

2. What is the effect of IODVA1 on receptor-stimulated activation of Rac? A simple Rac-GTP pull-down assay would address this question.

6. PLOS authors have the option to publish the peer review history of their article (what does this mean?). If published, this will include your full peer review and any attached files.

Reviewer #1: No

Reviewer #2: No

---

## [Author Response · Author response to Decision Letter 0]

3 Feb 2020

Response to Reviewers

I would like to thank the reviewers for their insightful comments. In general, both reviewers were enthusiastic to our findings and found minor weaknesses. By addressing the weaknesses pointed by the reviewers, I believe we improved on the manuscript’s quality and strengthened our conclusion.

Specifically, in response to reviewer 1 we tested the ability of the small molecule IODVA1 to inhibit the Rac-related small GTPases Cdc42 and Rho and found that IODVA1 is specific for Rac even though at highest tested concentration it decreases Cdc42’s activity (revised Fig 5C). Levels of active Rho were not affected by IODVA1. Similarly, we checked the activity of PAK4 downstream of Cdc42 and tested levels of pPAK4/5/6. PAK4/5/6 are specifically responsive to active Cdc42. Consistent with its weak activity on Cdc42, we found that levels of pPAK4/5/6 remain unchanged even after 3 h incubation with IODVA1 (revised Fig 5D). We also included new cell biology data (S4C Fig) that show that stress fibers, which are regulated by the activity of Rho, are not affected by IODVA1 consistent with the biochemical data. Together, these new data strengthen our conclusion that IODVA1 is specific to Rac but not to Cdc42 or Rho.

As requested by reviewer 2, we quantified all bands in presented immunoblots, e.g. revised Fig 2, and when appropriate, incorporated statistical analysis. In the figure legends., we added the number of times each experiment was repeated. The statistical analysis and quantification strengthened our conclusion that IODVA1 is not a Ras inhibitor.

We have also rearranged figures 2-6 to maximize clarity and inspected the final TIF files using PACE. We provide the data for Figure 6A, originally stated as “data not shown” as a supplemental figure (S4B Fig).

We checked for typos in the text and added references (e.g. Lu et al. 2017, ref#17 and Babagana et al., 2017 ref#18) describing the importance of the Rac/PAK pathway in inducing mechanisms of resistance to BRAF/MEK/ERK inhibitors in the clinic. We have also replaced the phrase ‘data not shown’ by actual data (e.g. S4B & S4C Fig).

Other specific changes to the text can be found in our response to the reviewers.

Reviewer #1:

We thank this reviewer for her/his insightful comments. By studying the effects of IODVA1 on Cdc42 and Rho activation and phosphorylation levels of PAK4/5/6, which were missing in the previous submission, we believe we have strengthened our paper. As this reviewer noted, the approach we followed in identifying IODVA1 is different from the ones used for other small GTPase inhibitors. We thought it is important to exactly report our approach given the difficulties in targeting protein-protein interactions especially ones involving small GTPases. Interestingly, our approach yielded a Rac inhibitor that is efficacious at the sub-micromolar concentration.

Specific comments:

1. Isogenic cell lines with different ras mutations would be better model to prove the specificity

We agree with this reviewer that using isogenic cell lines expressing different Ras mutations might have yielded more specific inhibitors. However, this is not the path we followed in this work and a Rac inhibitor is as interesting given the involvement of Rac in human cancer, in Ras transformation, and in resistance to current therapies. In addition, a Rac inhibitor with in vivo efficacy is still lacking.

2. Not clear how specificity will be achieved in cancer cells. RAS active conformation will also be needed for wt-KRAS function

Collectively our data is consistent with IODVA1 inhibiting Rac activation. We have shown that IODVA1 and parent NSC124205 do not inhibit Ras even though that was our intent.

3. The animal studies are not very convincing and the tumors seem to rebound to a size that is equivalent to control arm

We recognize the shortcomings of our animal experiments. We have set out to perform animal studies as a proof-of-concept, to establish if there is any potential of our small molecule inhibitor (SMI) and to observe any gross effects on animal health overall. The in vivo studies were done prior to maximum tolerated dose (MTD) and pharmacokinetic studies, which we are currently assessing. We also recognize that MDA-MB-231 is a triple negative breast cancer cell line and one of the most aggressive cancer cells lines to use in xenograft model, as it leads to formation of metastases in lymph nodes and lungs in 100% of NSG mice (Iorns et al., PLoS One 2012).

Additionally, we used an unoptimized dose of 3.5mg/kg every other day for all the animal studies, which is comparably lower to some other investigational SMIs. For example, an investigational drug motesanib, a VEGF antagonist, was used at 25 – 75 mg/kg twice daily in MDA-MB-231 xenograft (Coxon et al., Clin Cancer Res 2009) and Gefitinib, a FDA approved EGFR inhibitor, was used at 100 mg/kg orally in the same model (Liu et al., PLoS One 2017; Moon et al., Arch Pharm Res 2009). It is remarkable that IODVA1 at the low dose we used has any in vivo effect. The increase in tumor size we observe in both Ras-driven cancer models can be attributed to mechanisms of resistance. As we are improving our understanding of all the parameters of our drug, we will continue to optimize the drug delivery.

4. Rac pathway and downstream effector PAK1 cannot be studied in isolation. Other Group I and Group II PAKs need to be studied

Our primary motivation to look at the effects on PAK1/2 was that they are activated downstream of Rac and Cdc42, unlike PAK4/5/6, which are primarily under control of Cdc42. We have checked the effect of IODVA1 on class II PAKs (Fig 5D, right panel) and saw no appreciable difference in the phosphorylation, compared to PAK1/2 (Fig 5D, left panel). These results are consistent with apparent specificity of IODVA1 at the used concentration to Rac.

5. Besides RAC other RhoGTPases and their effectors need to be studied as well

We have additionally checked for effects on activation of Cdc42 and RhoA. We see a modest decrease in activation of Cdc42 and only at the highest IODVA1 concentration (Fig 5C, middle panel). We believe our SMI does not target RhoA as we saw no effect on the levels of active RhoA (Fig 5C, right panel) by pull-down and have not observed any changes in the stress fiber organization (Figure S4C), a well-known target of RhoA activity (Ridley and Hall, Cell 1992, Vexler et al., J Biol Chem 1996; Jatho et al., PLoS One 2015). Our early observations that IODVA1 is acting to inhibit lamellipodia and circular dorsal ruffle formation led us to posit that it inhibits Rac activation. Because Rac activation is required for formation of lamellipodia and circular dorsal ruffles and their loss cannot be rescued by Cdc42 or RhoG, we think the primary target of our SMI is activation of Rac (Nobes and Hall, Cell 1995; Ridley, Cell 2011; Krause and Gautreau, Nat Rev Mol Cell Biol 2014; Steffen et al., J Cell Sci 2013). The primary goal of this paper was to identify and synthesize the active ingredient in the chemical mixture and to pinpoint its potential mechanism of action. Now that we have established that IODVA1 targets activation of Rac, our immediate plans are to identify its target, which will be put forth in future studies.

6. PAK4 and its role in lamellipodia formation is known and the authors should evaluate the additional paks

We refer the reviewer to the previous point. IODVA1 does not affect PAK4 phosphorylation even following 3 h cell incubation. PAK4 was the first identified member of Class II PAKs and a Cdc42 interactor that leads to formation of filopodia (Abo et al., EMBO J 1998).

7. Background and introduction as well as discussion should include references on the different studies that have looked at RAC and PAK downstream of RAS post SMI treatment

Except for the G12C mutant, there are no small molecule inhibitors of Ras. As we mention in the introduction, most efforts have been focused on targeting its downstream effector kinases such as RAF, MEK and ERK. We have included additional references on most recent studies that discuss activation of Rac/PAK signaling axis in patients treated with RAF and MEK kinase inhibitors as mechanisms of resistance (Babagana et al., Mol Carcinog 2017; Lu et al., Nature 2017).

8. Typos should be checked and corrected

We thank the reviewer for pointing out typos. We have checked and corrected the document for typos.

9. Some of the figures need improvement in the style of presentation

We have rearranged the panels and improved the presentation of the figures.

Reviewer #2: 

We thank this reviewer for her/his enthusiasm and interest in our work and the way it was presented. The discovery of IODVA1 followed an unconventional path mixing targeted and phenotypic approaches and is distinct from the way other Ras or Rac inhibitors were identified. We thought reporting our approach should help others in the field especially that inhibiting protein-protein interactions is still difficult and that Rac and its regulators engage flat interfaces with no druggable cavities. To add another level of difficulty, the parent NCI compound was a mixture of compounds. Interestingly, IODVA1 is one of the few Rac inhibitors with in vivo activity against cancer mouse models.

1. The authors should provide a better quantitative analysis of data, specifically densitometry analysis of Western blots and indication of reproducibility/mean+/-S.E.

We have included the densitometry analysis for western blots (e.g. Fig 2) and included graphs quantifying signaling (e.g. Fig 4C, Fig 5C & 5D) and cell-based experiments (e.g. Fig S4B). We have also included relevant details in the figure legends. Additionally, key experiments were independently repeated by AG, GM and SH during the study without prior knowledge of the outcome.

2. What is the effect of IODVA1 on receptor-stimulated activation of Rac? A simple Rac-GTP pull-down assay would address this question.

Our main readout for activation of Rac is lamellipodia formation (Fig 5A) and formation of circular dorsal ruffles (Fig 5B), which are done under stimulatory conditions (activation of EGFR and PDGFR, respectively). Both cellular readouts are consistent with IODVA1 targeting the Rac signaling pathway. A pull-down assay with GST-PAK-GBD, which binds preferentially to the active form of Rac is provided in Fig 5C and shows that IODVA1 decreases Rac activity and that its action is specific to Rac but not Cdc42 or Rho. As we go on to further delineate the mechanism of action of IODVA1, we will certainly dissect its effect on Rac activation under receptor signaling.

3. Of course the main question that remains is how IODVA1 acts at a mechanistic level. This may be an effect on Rac or on Rac-GEF/Rac interactions,

We agree with this reviewer on his proposed target for IODVA1. In a series of experiments beyond the scope of this work, we are testing if IODVA1 interacts with Rac or if it activates the Rac-negative regulators p50GAP or RhoGDI1. Alternatively, it might bind to and inhibit the activation of one RacGEF thereby deactivating Rac.

---

## [Decision Letter · Decision Letter 1]

14 Feb 2020

IODVA1, a guanidinobenzimidazole derivative, targets Rac activity and Ras-driven cancer models.

PONE-D-19-30856R1

Dear Dr. Nassar,

We are pleased to inform you that your manuscript has been judged scientifically suitable for publication and will be formally accepted for publication once it complies with all outstanding technical requirements.

With kind regards,

Irina V. Lebedeva, Ph.D.

Academic Editor

PLOS ONE

Additional Editor Comments (optional):

Reviewers' comments:

Reviewer's Responses to Questions

**Comments to the Author**

1. If the authors have adequately addressed your comments raised in a previous round of review and you feel that this manuscript is now acceptable for publication, you may indicate that here to bypass the “Comments to the Author” section, enter your conflict of interest statement in the “Confidential to Editor” section, and submit your "Accept" recommendation.

Reviewer #1: All comments have been addressed

Reviewer #2: All comments have been addressed

2. Is the manuscript technically sound, and do the data support the conclusions?

Reviewer #1: Yes

Reviewer #2: Yes

3. Has the statistical analysis been performed appropriately and rigorously? 

Reviewer #1: Yes

Reviewer #2: Yes

4. Have the authors made all data underlying the findings in their manuscript fully available?

Reviewer #1: Yes

Reviewer #2: Yes

5. Is the manuscript presented in an intelligible fashion and written in standard English?

Reviewer #1: Yes

Reviewer #2: Yes

6. Review Comments to the Author

Reviewer #1: Comments have been addressed

The manuscript in in a much better shape

Figures are also well organized and additional data supports the overall hypothesis.

Reviewer #2: Authors addressed all comments. This is a very interesting manuscript, highly appropriate for PLoS One.

7. PLOS authors have the option to publish the peer review history of their article (what does this mean?). If published, this will include your full peer review and any attached files.

Reviewer #1: No

Reviewer #2: No

---

## [Editor Report · Acceptance letter]

19 Feb 2020

PONE-D-19-30856R1 

IODVA1, a guanidinobenzimidazole derivative, targets Rac activity and Ras-driven cancer models. 

Dear Dr. Nassar:

I am pleased to inform you that your manuscript has been deemed suitable for publication in PLOS ONE. Congratulations! Your manuscript is now with our production department. 

With kind regards,

on behalf of

Dr. Irina V. Lebedeva 

Academic Editor

PLOS ONE